# Weakly migratory metastatic breast cancer cells activate fibroblasts via microvesicle-Tg2 to facilitate dissemination and metastasis

Samantha C Schwager[1], Katherine M Young[1], Lauren A Hapach[1,2], Caroline M Carlson[1], Jenna A Mosier[1], Tanner J McArdle[3], Wenjun Wang[1], Curtis Schunk[1], Anissa L Jayathilake[4], Madison E Bates[1], Francois Bordeleau[5], Marc A Antonyak[6], Richard A Cerione[6], Cynthia A Reinhart-King[7]*

[1]Department of Biomedical Engineering, Vanderbilt University, Nashville, United States; [2]Department of Biomedical Engineering, Cornell University, Ithaca, United States; [3]Vanderbilt University Medical Center, Nashville, United States; [4]Hume-Fogg Academic High School, Nashville, United States; [5]CHU de Québec-Université Laval Research Center (Oncology division), UniversitéLaval Cancer Research Center and Faculty of Medicine, Université Laval, Québecc, Canada; [6]Department of Biomedical Science, Cornell University, Ithaca, United States; [7]Department of Chemistry and Chemical Biology, Cornell University, Ithaca, United States

*For correspondence: Cynthia.Reinhart-King@vanderbilt.edu

Competing interest: The authors declare that no competing interests exist.

**Abstract** Cancer cell migration is highly heterogeneous, and the migratory capability of cancer cells is thought to be an indicator of metastatic potential. It is becoming clear that a cancer cell does not have to be inherently migratory to metastasize, with weakly migratory cancer cells often found to be highly metastatic. However, the mechanism through which weakly migratory cells escape from the primary tumor remains unclear. Here, utilizing phenotypically sorted highly and weakly migratory human breast cancer cells, we demonstrate that weakly migratory metastatic cells disseminate from the primary tumor via communication with stromal cells. While highly migratory cells are capable of single cell migration, weakly migratory cells rely on cell-cell signaling with fibroblasts to escape the primary tumor. Weakly migratory cells release microvesicles rich in tissue transglutaminase 2 (Tg2) which activate murine fibroblasts and lead weakly migratory cancer cell migration in vitro. These microvesicles also induce tumor stiffening and fibroblast activation in vivo and enhance the metastasis of weakly migratory cells. Our results identify microvesicles and Tg2 as potential therapeutic targets for metastasis and reveal a novel aspect of the metastatic cascade in which weakly migratory cells release microvesicles which activate fibroblasts to enhance cancer cell dissemination.

## Editor's evaluation

This is an important study demonstrating that poorly migratory breast cancer cells can be metastatic by activating fibroblasts via Tg2-containing microvesicles (MVs). A convincing array of methodologies reveal the metastasis-promoting qualities of MV-associated Tg2 and also demonstrates that cancer cells can communicate with the microenvironment in order to overcome tumour-cell intrinsic deficiencies in migratory capacity. This work will be of interest to cancer biologists studying tumour heterogeneity and the role of the tumour microenvironment in metastatic progression.

## Introduction

It is known that cancer cell migration in vivo can be highly heterogeneous with cells exhibiting a wide range of migratory phenotypes, including amoeboid, mesenchymal, single cell, and collective migration (*Clark and Vignjevic, 2015*; *Aiello et al., 2018*; *De Pascalis and Etienne-Manneville, 2017*). Even within a single population, cells can exhibit different migration phenotypes (*Hapach et al., 2021*; *Kwon et al., 2019*; *Summerbell et al., 2020*; *Hallou et al., 2017*) resulting from intrinsic cancer cell genetic differences (*Dai et al., 2017*; *Wu et al., 2008*; *Patsialou et al., 2012*; *Walkiewicz et al., 2016*) and extrinsic factors, such as interactions with the extracellular matrix or stromal cells (*Clark and Vignjevic, 2015*; *Mosier et al., 2019*; *Zanotelli et al., 2019*; *Attieh et al., 2017*; *Fuchigami et al., 2017*; *Gaggioli et al., 2007*; *Labernadie et al., 2017*). As the first step in the metastatic cascade involves the migration and invasion of cancer cells away from the primary tumor, migratory capability is largely believed to be an indicator of cancer progression (*van Zijl et al., 2011*; *Palmer et al., 2011*; *Fares et al., 2020*; *Liu et al., 2020*). However, it is now becoming clear that a cancer cell does not have to be inherently migratory to metastasize (*Hapach et al., 2021*; *Chen et al., 2017*; *Fietz et al., 2017*; *Johnstone et al., 2018*; *Padmanaban et al., 2019*; *Kubens and Zänker, 1998*). In fact, some of the highly metastatic phenotypes in breast and colorectal cancers are often less migratory than the weakly metastatic phenotypes, yet they are still able to enter the circulation efficiently (*Hapach et al., 2021*; *Fietz et al., 2017*; *Padmanaban et al., 2019*; *Kubens and Zänker, 1998*; *Tormoen et al., 2012*). While enhanced clustering, survival, and proliferation have been suggested as potential mechanisms for why weakly migratory cells can outperform their highly migratory counterparts in the late stages of the metastatic cascade thus contributing to metastasis (*Hapach et al., 2021*; *Padmanaban et al., 2019*), it is not clear how these weakly migratory cells can efficiently escape from the primary site.

To escape the primary site during local invasion, cancer cells navigate through a heterogeneous tumor microenvironment, where they interact with the extracellular matrix and a diverse collection of stromal cells. Chemical, physical, and metabolic interactions with the tumor microenvironment are known to alter the invasion capacity of cancer cells (*Attieh et al., 2017*; *Zanotelli et al., 2021*; *Zhou et al., 2020*). Cancer associated fibroblasts (CAFs) are a stromal cell in the tumor microenvironment that can lead and promote tumor cell invasion and metastasis through extracellular matrix (ECM) remodeling (*Attieh et al., 2017*; *Fuchigami et al., 2017*; *Miyazaki et al., 2020*; *Provenzano et al., 2006*; *Erdogan and Webb, 2017*; *Dumont et al., 2013*; *Goetz et al., 2011*). CAFs at the primary tumor are largely derived from fibroblasts that have been transformed to a more contractile, activated state (*Kalluri, 2016*), and cancer-derived microvesicles (MVs) have recently been implicated in fibroblast activation (*Antonyak et al., 2011*; *Schwager et al., 2019*; *Jiang et al., 2019*). Thus, we hypothesized that while highly migratory cells can escape the primary tumor independently, CAFs facilitate weakly migratory cancer cell escape from the primary tumor.

Here, we identify a novel aspect of the metastatic cascade by which weakly migratory cancer cells release tissue transglutaminase 2 (Tg2)-rich MVs to activate fibroblasts and enhance cancer cell dissemination. Moreover, highly and weakly migratory cells release MVs which differentially signal to the tumor microenvironment. MVs from highly migratory cells have little effect on fibroblast activation, and highly migratory cells do not require activated fibroblasts to migrate. In contrast, MVs from weakly migratory cells are rich in Tg2 and activate fibroblasts which enhance fibroblast-led weakly migratory cancer cell migration in vitro. These MVs also induce tumor stiffening and fibroblast activation in vivo and enhance the dissemination and metastasis of weakly migratory cells. Our findings highlight MVs and Tg2 as potential targets for developing therapeutics to prevent metastasis.

## Results

### Highly and weakly migratory breast cancer subpopulations form tumors with distinct matrix and fibroblast populations

To investigate the mechanism through which weakly migratory cells locally invade and disseminate, MDA-MB-231 breast cancer cells were phenotypically sorted based on migration through a collagen coated transwell (*Hapach et al., 2021*). Twenty rounds of sorting resulted in the isolation of two stable subpopulations of breast cancer cells: highly migratory MDA⁺ and weakly migratory MDA⁻ (*Figure 1a*). Consistent with our previous in vivo findings (*Hapach et al., 2021*), the highly migratory MDA⁺ subpopulation and the weakly migratory MDA⁻ subpopulation formed orthotopic primary tumors that

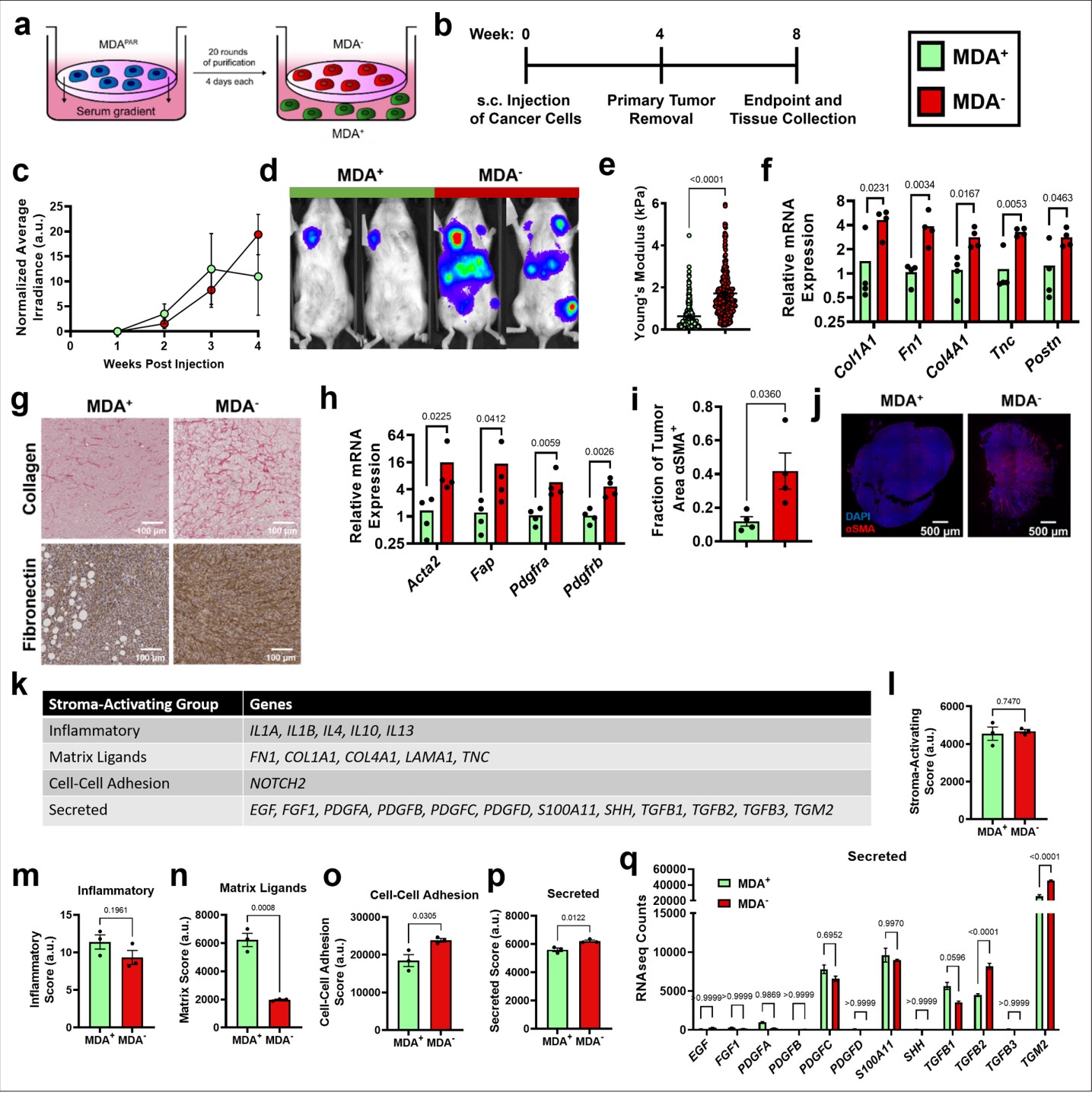

**Figure 1.** Highly and weakly migratory breast cancer subpopulations form tumors with distinct matrix and fibroblast populations. (**a**) Schematic of phenotypic sorting. (**b**) Timeline of orthotopic mammary metastasis model. (**c**) Normalized average irradiance of MDA+ and MDA- primary tumors (N=3 mice). (**d**) Representative BLI of MDA+ and MDA- metastasis. (**e**) AFM stiffness measurements of MDA+ and MDA- primary tumors. (N=3 + tumors per condition; n=254, 280). (**f**) Relative mRNA expression of mouse-derived matrix components in MDA+ and MDA- primary tumors. (N=4 tumors per condition). (**g**) Immunohistochemical staining of collagen and fibronectin in MDA+ and MDA- primary tumors. (**h**) Relative mRNA expression of stromal CAF markers in MDA+ and MDA-. (N=4 tumors per condition). (**i**) Fraction of tumor area positive for αSMA (N=4). (**j**) Representative images of αSMA (red) and DAPI (blue) in MDA+ and MDA- tumors. (**k**) Stroma-Activating Score Group genes identified in RNAseq of MDA+ and MDA-. (**l**) Stroma-Activating Score of MDA+ and MDA- (N=3). (**m**) Inflammatory Score of MDA+ and MDA- (N=3). (**n**) Matrix Score of MDA+ and MDA- (N=3). (**o**) Cell-Cell Adhesion Score of MDA+ and MDA- (N=3). (**p**) Secreted Score of MDA+ and MDA- (N=3). (**q**) Inflammatory Score of MDA+ and MDA- (N=3). Secreted Score of MDA+ and MDA-, separated by gene (N=3). mRNA graphs show mean + individual data points. Bar graphs shown mean +/- SEM. p-values determined using an unpaired Student's t-test. Source data available in *Source data 2*.

grew at comparable rates (*Figure 1b–c*) and the MDA$^+$ subpopulation was weakly metastatic while the MDA$^-$ subpopulation was highly metastatic in vivo (*Figure 1d*). Given the differing migratory and metastatic capabilities of MDA$^+$ and MDA$^-$, we sought to characterize the tumor microenvironment formed by highly and weakly migratory cells in vivo to determine the impact of the tumor microenvironment on cancer cell dissemination and subsequent metastasis.

Since increased collagen and fibronectin deposition and increased tumor stiffness have been linked to tumor progression (*Piersma et al., 2020*; *Fenner et al., 2014*; *Provenzano et al., 2008*), we investigated the mechanical properties of MDA$^+$ and MDA$^-$ tumors to assess whether altered tumor matrix mechanics may contribute to metastatic potential. Based on atomic force microscopy measurements, MDA$^-$ primary tumors exhibited significantly higher tumor stiffness compared to MDA$^+$ tumors (*Figure 1e*). qPCR of tumors revealed significantly higher amounts of mouse-derived matrix, including collagen I (*Col1a1*), fibronectin 1 (*Fn1*), collagen IV (*Col4a1*), tenascin-C (*Tnc*), and periostin (*Postn*), in MDA$^-$ tumors compared to MDA$^+$ tumors (*Figure 1f*). Similarly, increased collagen and fibronectin was evident via immunohistochemical staining in MDA$^-$ tumors compared to MDA$^+$ tumors (*Figure 1g*). These results indicate that the weakly migratory, highly metastatic MDA$^-$ form stiffer tumors with increased ECM compared to the highly migratory, weakly metastatic MDA$^+$.

Given that MDA$^-$ produce stiffer tumors than MDA$^+$, and CAFs are a major stromal component of breast tumors that mediate matrix deposition (*Liu et al., 2019*), we investigated the CAF component of tumors from each subpopulation. MDA$^-$ tumors exhibited increased levels of several CAF markers, including α-smooth muscle actin (*Acta2*), fibroblast activation protein (*Fap*), and platelet-derived growth factor receptors α and β (*Pdgfra*, *Pdgfrb*) (*Figure 1h*). MDA$^-$ tumors also exhibited an increased fraction of αSMA positive tissue area compared to MDA$^+$ tumors (*Figure 1i-j*). Given that αSMA is the primary marker of myofibroblast-like CAFs (myCAFs), that myCAFs and matrix-CAFs (mCAFs) express high levels of Fap, *Pdgfra*, and *Pdgfrb* (*Öhlund et al., 2017*; *Wu et al., 2020*; *Bartoschek et al., 2018*), and that myCAFs are highly contractile and deposit high levels of ECM (*Wu et al., 2020*; *Kieffer et al., 2020*), these results suggest that MDA$^-$ tumors are enriched for a myCAF-like fibroblast population. These findings reveal that the weakly migratory, highly metastatic MDA$^-$ primary tumors have increased myCAF-like fibroblasts.

Since MDA$^-$ tumors are stiffer and have a larger population of CAFs compared to MDA$^+$ tumors, we investigated the mechanisms by which MDA$^-$ may activate fibroblasts using RNA-seq. Previously published genes involved in cancer cell-induced fibroblast activation (*Merlino et al., 2016*; *Belhabib et al., 2021*; *Wu et al., 2021*; *Linares et al., 2020*) were used to generate a Stroma-Activating Score based on RNA-seq of MDA$^+$ and MDA$^-$ (*Figure 1k–l*). No significant difference in Stroma-Activating Score was calculated between MDA$^+$ and MDA$^-$ (*Figure 1l*). However, when separated into several Stroma-Activating Groups (Inflammatory, Matrix Ligands, Cell-Cell Adhesion, and Secreted), MDA$^+$ scored significantly higher on Matrix Ligands, while MDA$^-$ scored significantly higher on Cell-Cell Adhesion and Secreted genes (*Figure 1m–p*). To further investigate the secreted factors that may influence fibroblast activation, RNA-seq counts of genes involved in fibroblast activation through secreted factors was assessed (*Figure 1q*). Both *TGFB2* and *TGM2* were significantly enriched in MDA$^-$ compared to MDA$^+$. These findings suggest that MDA$^-$ activate fibroblasts through secreted factors.

## MVs released from weakly migratory cancer cells are potent activators of fibroblasts in vitro

Since our data indicate that MDA$^-$ may activate fibroblasts using secreted factors, and it is known that MVs can induce fibroblast activation (*Antonyak et al., 2011*; *Schwager et al., 2019*), we compared the MVs released from MDA$^+$ and MDA$^-$. MVs were collected from the MDA$^+$ and MDA$^-$ subpopulations after culture in serum-free conditions. Isolated MVs were purified using centrifugation and size-based filtration, and were termed MV$^+$ and MV$^-$, respectively. MDA$^-$ released significantly more MVs than MDA$^+$ (*Figure 2a*), but both MV$^+$ and MV$^-$ had similar size distributions and were within the expected 200 nm – 1 µm range (*Figure 2b*). To probe MV signaling to fibroblasts, fibroblast phenotypes associated with fibroblast activation and cancer progression were examined after culture with MV$^+$ and MV$^-$ on 20 kPa polyacrylamide gels, representing the stiffness of breast ECM at the tumor periphery (*Plodinec et al., 2012*; *Figure 2c*). MV$^-$ caused focal adhesion kinase (FAK) Tyr397 phosphorylation in fibroblasts, as increased FAK phosphorylation (pFAK) was evident in fibroblasts cultured with MV$^-$ compared to control and MV$^+$ conditions (*Figure 2d–e*), consistent with previous findings that

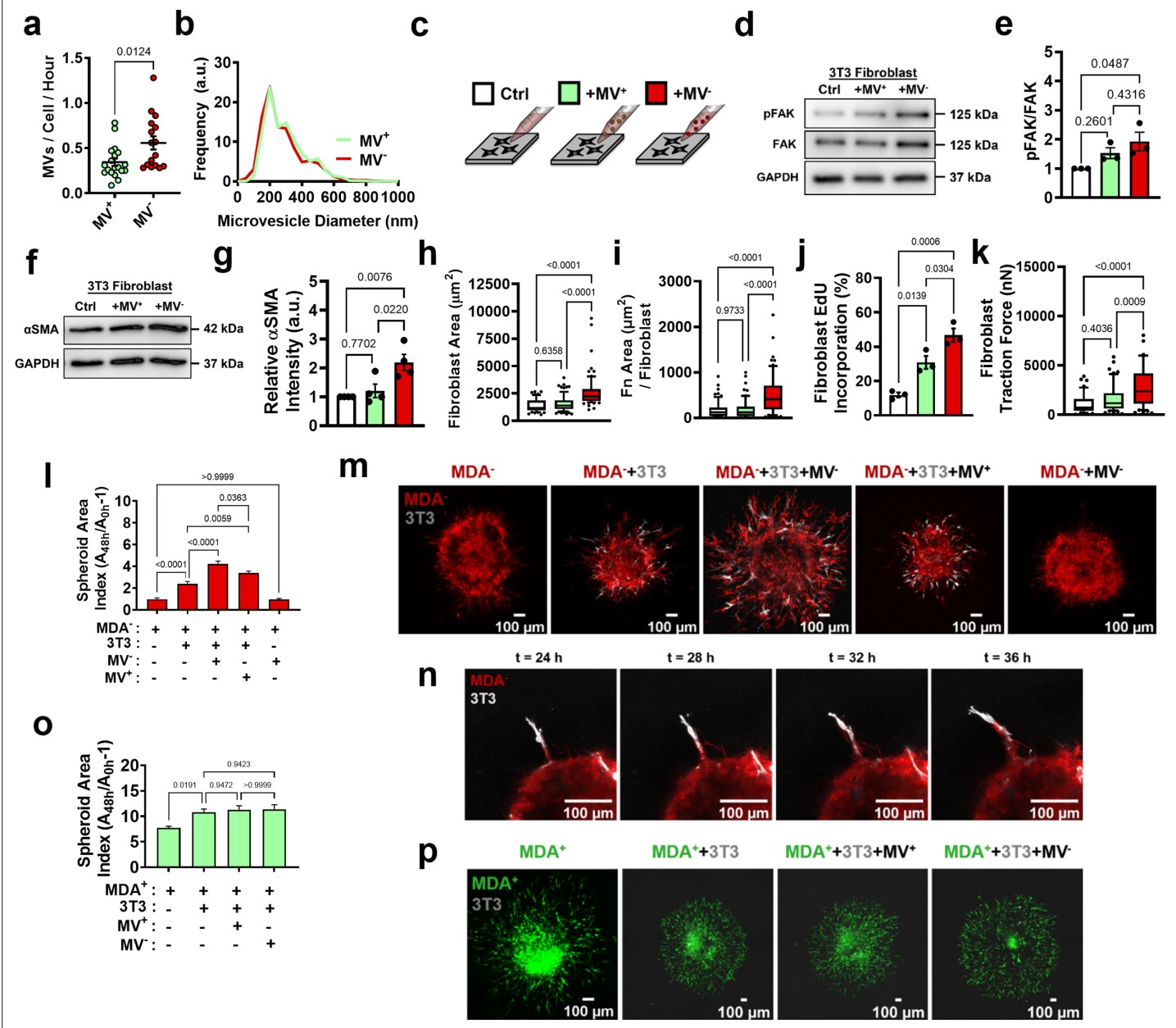

**Figure 2.** MV⁻ are potent activators of fibroblasts in vitro. (**a**) Number of MVs released from MDA⁺ and MDA⁻ per hour (N=19, 16). (**b**) Size distribution of MVs isolated from MDA⁺ and MDA⁻. (**c**) Schematic overviewing fibroblast culture with MVs. (**d**) Western blot of pFAK, FAK, and GAPDH in fibroblasts cultured in cultured in control conditions (Ctrl), with MV⁺ (+MV⁺), or with MV⁻ (+MV⁻). (**e**) Quantification of western blot from (**d**) (N=3). (**f**) Western blot of αSMA and GAPDH in fibroblasts cultured in in Ctrl, +MV⁺, or +MV⁻ conditions (**g**) Quantification of western blot from (**f**) (N=4). (**h**) Fibroblast cell area after culture in Ctrl, +MV⁺, or +MV⁻ conditions. (N=3; n=69, 75, 77). (**i**) Area of fibronectin deposited per fibroblast cultured in Ctrl, +MV⁺, or +MV⁻ conditions. (N=3; n=66, 66, 61). (**j**) Percentage of fibroblasts EdU positive after culture in Ctrl, +MV⁺, or +MV⁻ conditions. (N=3). (**k**) Fibroblast traction force after culture in Ctrl, +MV⁺, or +MV⁻ conditions. (N=3; n=41, 57, 58). (**l**) Representative images of spheroid outgrowth 48 hr post embedding. MDA⁻ (red); 3T3 fibroblast (gray). (**m**) Spheroid area index 48 hr post embedding. (N=3 + ; n=22, 31, 32, 22, 19). (**n**) Time series images of fibroblast leading MDA⁻ escape from spheroid. (**o**) Spheroid area index 48 hr post embedding. (N=3 + ; n=28, 62, 34, 28). (**p**) Representative images of spheroid outgrowth 48 hr post embedding. MDA⁺ (green); 3T3 fibroblast (gray). Bar graphs show mean +/- SEM. Box and whisker plots show median and 25th-75th (box) and 10th-90th (whiskers) percentiles. p-Values determined using an unpaired Student's t-test or a one-way ANOVA with Tukey's test for multiple comparisons. Source data available in **Source data 1 and 2**.

The online version of this article includes the following figure supplement(s) for figure 2:

**Figure supplement 1.** MV⁻ activate fibroblasts in vitro.

MDA-MB-231 MVs regulate fibroblast FAK activation (*Antonyak et al., 2011*). Additionally, fibroblasts cultured with MV$^-$ exhibited increased α-smooth muscle actin (αSMA) expression, a marker of fibroblast activation, and cell area compared to both control and MV$^+$ conditions (*Figure 2f–h*, *Figure 2—figure supplement 1a*). Fibroblasts cultured with MV$^+$ displayed no change in αSMA expression or cell area compared to control conditions (*Figure 2f–h*, *Figure 2—figure supplement 1a*). Fibroblasts cultured with MV$^-$ also displayed increased positive fibronectin staining (*Figure 2i*, *Figure 2—figure supplement 1b–c*), suggesting the MV$^-$ increased fibronectin deposition by fibroblasts. Fibroblasts cultured with MV$^-$ exhibited increased EdU incorporation compared to MV$^+$ and control conditions, indicative of increased proliferation (*Figure 2j*, *Figure 2—figure supplement 1d*). Increased traction force and traction stress was evident in fibroblasts cultured with MV$^-$, compared to control conditions, but not in fibroblasts cultured with MV$^+$ (*Figure 2k*, *Figure 2—figure supplement 1e–f*). Overall, this data indicates that MV$^-$ activate fibroblasts to a more proliferative, contractile state compared to MV$^+$ and control conditions.

## MV-mediated fibroblast activation increases weakly migratory cancer cell migration

CAFs enable cancer cell migration and metastasis, in part, through matrix remodeling at the primary tumor (*Attieh et al., 2017*; *Fuchigami et al., 2017*; *Miyazaki et al., 2020*; *Provenzano et al., 2006*; *Erdogan and Webb, 2017*; *Dumont et al., 2013*; *Goetz et al., 2011*). Given that our data indicate that MDA$^-$ cells induce fibroblast activation through the release of MV, we investigated whether MV-activated fibroblasts could promote the migration of MDA$^-$ in a tumor spheroid model. After 48 hr of culture, MDA$^-$ spheroids exhibited minimal spheroid outgrowth, measured using a spheroid area index ($A_{48h}/A_{0h}$-1) (*Figure 2l–m*, *Figure 2—figure supplement 1g–h*), while MDA$^+$ spheroids exhibited high levels of spheroid outgrowth (*Figure 2o–p*, *Figure 2—figure supplement 1g and j*), consistent with our previous findings (*Hapach et al., 2021*). Coculture of MDA$^-$ or MDA$^+$ with 3T3 fibroblasts (MDA$^-$+3 T3, MDA$^+$+3 T3) resulted in significantly enhanced spheroid outgrowth compared to cancer cells alone (*Figure 2l–m and o–p*, *Figure 2—figure supplement 1g–h and j*). MDA$^-$+3 T3 spheroids displayed fibroblast-led strands of MDA$^-$ migration away from the spheroid, suggesting that the fibroblasts physically lead the migration of MDA$^-$ through the matrix (*Figure 2n*). Importantly, MDA$^-$+3 T3 spheroids cultured with MV$^-$ exhibited significantly enhanced spheroid outgrowth and MDA$^-$ migration distance, compared to MDA$^-$+3 T3 control spheroids (*Figure 2l–m*, *Figure 2—figure supplement 1g–h*). MDA$^-$+3 T3 spheroids cultured with MV$^+$ also showed increased outgrowth compared to MDA$^-$+3 T3 spheroids, but this outgrowth was significantly less than that induced by MV$^-$ (*Figure 2l–m*, *Figure 2—figure supplement 1g–h*). Interestingly, MDA$^+$+3 T3 spheroids cultured with MV$^+$ or MV$^-$ exhibited no change in spheroid outgrowth and MDA$^+$ migration distance compared to control conditions (*Figure 2o–p*, *Figure 2—figure supplement 1g and j*). This result suggests that MDA$^+$ are capable of robust migration independently of MV-mediated fibroblast signaling. When MV$^-$ were applied to MDA$^-$ only spheroids, no increase in spheroid outgrowth was observed, highlighting the necessity of fibroblasts for this MV-induced cancer cell migration (*Figure 2l–m*, *Figure 2—figure supplement 1g–h*). Additionally, MDA$^-$ spheroids cultured with fibroblast conditioned media or MV$^-$-treated fibroblast conditioned media displayed no change in spheroid outgrowth (*Figure 2—figure supplement 1i*), indicating that fibroblast-enhanced cancer cell migration is not a result of secreted factors but rather due to physical interactions. Altogether, these results indicate that MV$^-$ are potent activators of fibroblasts and that MV$^-$-induced fibroblast activation mechanically feeds back to induce the migration of MDA$^-$.

## Highly and weakly migratory breast cancer cells release MVs with distinctly different contents

To determine the MV cargo responsible for the activation of fibroblasts by the MDA$^-$ cells, iTRAQ proteomics was completed on MV$^+$ and MV$^-$. Identified proteins were compared to Exocarta and Vesiclepedia EV databases and exhibited significant overlap with both EV databases (*Figure 3—figure supplement 1a*). Proteomics data suggests that MV cargo was different between highly migratory and weakly migratory subpopulations of breast cancer cells, with 26.8% of the identified proteins being more highly expressed by MV$^+$ (FC ≥1.1) and 29.8% being more highly expressed by MV$^-$ (FC ≤0.9) (*Figure 3a*). 43.4% of identified proteins were similarly expressed between both

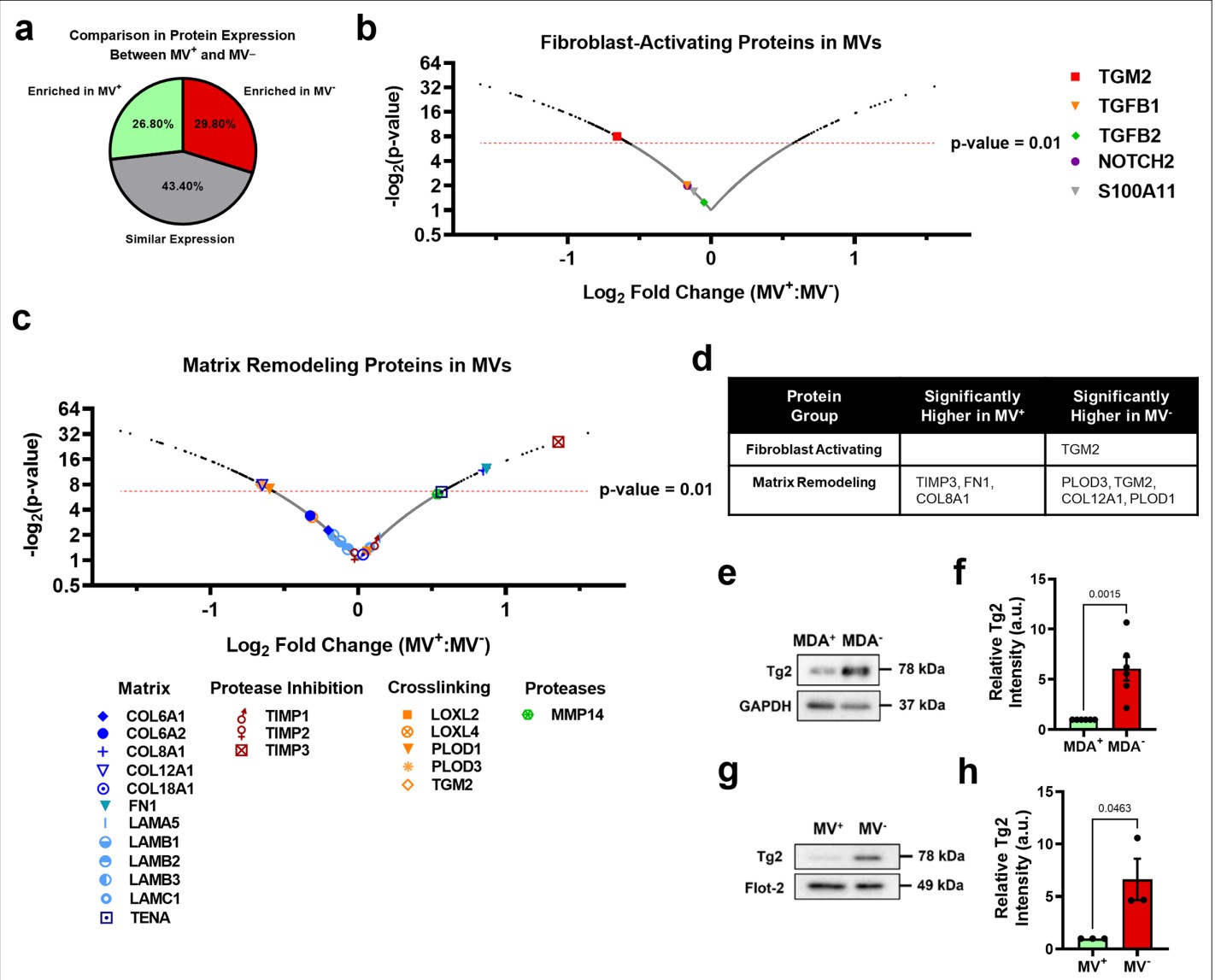

**Figure 3.** Phenotypically sorted breast cancer subpopulations release MVs with distinctly different contents. (**a**) Comparison of protein expression between MV⁺ and MV⁻. Enriched in MV⁺=FC ≥ 1.1; similar expression = 0.9 < FC<1.1; enriched in MV⁻=FC ≤ 0.9. (**b**) Fibroblast-activating proteins identified in proteomics of MV⁺ and MV⁻. p-Value cut-off of 0.01 indicated with red-dotted line. (**c**) Matrix remodeling proteins identified in in proteomics of MV⁺ and MV⁻. p-Value cut-off of 0.01 indicated with red-dotted line. (**d**) Significantly upregulated genes (P<0.01) in MV⁺ and MV⁻ involved in fibroblast activation and matrix remodeling. (**e**) Western blot of Tg2 and GAPDH in MDA⁺ and MDA⁻. (**f**) Quantification of western blot in (**e**) (N=6). (**g**) Western blot of Tg2 and Flot-2 in MV⁺ and MV⁻. (**h**) Quantification of western blot in (**g**) (N=3). Proteomics graphs show log₂ Fold Change of MV⁺:MV⁻ and each data point represents a protein. Bar graphs show mean +/- SEM. p-Values of bar graphs determined using an unpaired Student's t-test. Source data avilable in *Source data 1 and 2*.

The online version of this article includes the following figure supplement(s) for figure 3:

**Figure supplement 1.** Proteomic analysis of MVs released from phenotypically sorted breast cancer cells.

MV populations (0.9<FC < 1.1) (*Figure 3a*). Protein set enrichment analysis (PSEA) of MV⁺ and MV⁻ cargo was performed using the PSEA-Quant algorithm (*Lavallée-Adam et al., 2014*). PSEA revealed different GO annotations for MV⁺ and MV⁻ protein expression (*Figure 3—figure supplement 1b–c*). Of note, MV⁺ were enriched for cargo involved in biological adhesion, cell adhesion, location, and cell motility (*Figure 3—figure supplement 1b*). MV⁻ were enriched for cargo involved in various metabolic and catabolic processes and vesicle-mediated transport (*Figure 3—figure supplement 1c*). These findings indicate that cancer cells with varying migratory ability release MVs with distinctly different cargo. Additionally, RNA-seq of highly and weakly migratory MDA-MB-231 and proteomics of MVs

released by highly and weakly migratory MDA-MB-231 were compared to determine if protein differences in MVs were related to gene expression differences in the MV-releasing cell. 79% of genes/proteins identified in both datasets (1786 of 2260) were expressed at similar levels between cells and MVs (*Figure 3—figure supplement 1d*). 18.2% of genes/proteins were significantly different between cell populations, but not MV populations (*Figure 3—figure supplement 1d*). 3% of genes/proteins were significantly different between MV populations, but not cell populations. (*Figure 3—figure supplement 1d*). 3.5% of genes/proteins were significantly enriched in both cell and MV populations (*Figure 3—figure supplement 1d*). These results highlight that the majority of genes/proteins identified by RNA-seq and proteomics are expressed at similar levels between the MDA subpopulations and their respective MVs, reinforcing the notion that MVs reflect the composition of the MV-releasing cell.

Expression of proteins involved in fibroblast activation and matrix remodeling in MVs were assessed using proteomics (*Figure 3b–c*). The fibroblast-activating proteins TGM2, TGFB1, TGFB2, NOTCH2, and S100A11 were identified in MV$^+$ and MV$^-$ (*Figure 3b*). MV$^-$ had significantly higher expression of TGM2 compared to MV$^+$, while TGFB1, TGFB2, NOTCH2, and S100A11 had no significant difference in expression between MV$^+$ and MV$^-$ (*Figure 3b and d*). Many matrix-remodeling proteins were identified in MV$^+$ and MV$^-$, including a variety of matrix ligands, protease inhibitors, crosslinking enzymes, and matrix-degrading proteases (*Figure 3c*). TIMP3, FN1, and COL8A1 matrix-remodeling proteins were expressed significantly higher in MV$^+$ while PLOD3, TGM2, COL12A1, and PLOD1 were expressed significantly higher in MV$^-$ (*Figure 3c–d*). Tissue transglutaminase 2, also known as TGM2 or Tg2, identified as significantly enriched in MV$^-$ compared to MV$^+$, is a calcium-dependent enzyme that both crosslinks collagen and activates fibroblasts (*Figure 3d*). EVs have previously been shown to transfer Tg2 to fibroblasts resulting in fibroblast activation (*Antonyak et al., 2011*; *Shinde et al., 2020*). Increased Tg2 expression in MDA$^-$ and MV$^-$, compared to MDA$^+$ and MV$^+$, was verified using western blot (*Figure 3e–h*). It has also been shown that EV-Tg2 crosslinks EV-fibronectin (Fn) into dimers to potentiate fibroblast integrin signaling upon EV-fibroblast interactions (*Antonyak et al., 2011*; *Shinde et al., 2020*). While the Fn dimer was present in both MV$^+$ and MV$^-$, it was more highly expressed in MV$^+$ (*Figure 3—figure supplement 1e*), suggesting that fibroblast activation by MV$^-$ does not solely rely on Fn dimer-mediated integrin signaling.

## Modulation of MV-Tg2 expression regulates MV-mediated fibroblast activation and cancer cell dissemination in vitro

Given that Tg2 is a known mediator of MV-induced fibroblast transformation (*Antonyak et al., 2011*), we investigated whether knockdown of Tg2 abrogates MV$^-$-induced fibroblast activation. MDA$^-$ was stably transduced with shRNA targeting Tg2 (MDA$^{- (shTg2)}$) (*Figure 4—figure supplement 1a–b*). Knockdown of Tg2 resulted in no change in MDA$^-$ migration or MV release compared to the scrambled control (MDA$^{- (scr)}$) (*Figure 4—figure supplement 1c–d*). A slight decrease in transwell invasion was observed in MDA$^{- (shTg2)}$ compared to MDA$^{- (scr)}$ (*Figure 4—figure supplement 1e*). MVs released from MDA$^{- (shTg2)}$ (MV$^{- (shTg2)}$) had reduced Tg2 expression compared to the scrambled control MVs (MV$^{- (scr)}$) (*Figure 4—figure supplement 1f–g*).

Proteomics analysis of MV$^{- (scr)}$ and MV$^{- (shTg2)}$ was completed to identify other MV proteins that significantly changed with Tg2 knockdown that may affect MV-fibroblast signaling. 50.05% of proteins identified using proteomics were detected at similar levels between MV$^{- (scr)}$ and MV$^{- (shTg2)}$, 25.5% of proteins were enriched in MV$^{- (scr)}$, and 24.45% of proteins were enriched in MV$^{- (shTg2)}$ (*Figure 4—figure supplement 1h*). Protein set enrichment analysis (PSEA) of MV$^{- (scr)}$ and MV$^{- (shTg2)}$ cargo was performed using the PSEA-Quant algorithm (*Lavallée-Adam et al., 2014*). PSEA revealed different GO annotations for MV$^{- (scr)}$ and MV$^{- (shTg2)}$ protein expression (*Figure 4—figure supplement 1i–j*). MV$^{- (scr)}$ were enriched for cargo involved in response to stimuli, vesicle-mediated transport, and secretion (*Figure 4—figure supplement 1i*). MV$^{- (shTg2)}$ were enriched for cargo involved in a variety of metabolic processes and cell cycle (*Figure 4—figure supplement 1j*). These findings indicate that knockdown of Tg2 affected a variety of cell processes and resulted in different MV protein expression. Proteomics data was additionally analyzed to determine whether additional proteins involved in fibroblast activation and matrix remodeling were altered in MV$^{- (shTg2)}$. As expected, Tg2 was significantly enriched in MV$^{- (scr)}$ compared to MV$^{- (shTg2)}$ (*Figure 4—figure supplement 1k*). S100A11 was also significantly enriched in MV$^{- (scr)}$ compared to MV$^{- (shTg2)}$ (*Figure 4—figure supplement 1k*). However, S100A11 was

detected in both populations at low levels with only 6 peptides identified by proteomics compared to 24 peptides for Tg2. Importantly, S100A11 released by cancer cells is known to stimulate a change in fibroblast phenotype upon binding to the RAGE receptor (*Mitsui et al., 2019*). Dimerization of S100A11 by Tg2 is required for S100A11-RAGE binding (*Zhang et al., 2021*). Therefore, any effects on fibroblast activation due to an upregulation of MV S100A11 is likely due to Tg2 signaling. All other fibroblast-activating and matrix-remodeling proteins in MVs identified with proteomics were not significant altered by Tg2 knockdown (*Figure 4—figure supplement 1k*).

Fibroblasts cultured with MV$^{- (shTg2)}$ exhibited decreased αSMA expression, cell area, fibronectin area, proliferation, and traction force, compared to culture with MV$^{- (scr)}$ (*Figure 4a–f*, *Figure 4—figure supplement 2a–g*). This suggests that knockdown of Tg2 in MDA- cells reduced the capacity of MV$^-$ to activate fibroblasts. Treatment of MV$^-$ with the Tg2 inhibitor, T101, decreased fibroblast αSMA expression compared to culture with MV$^-$, further supporting role of MV-Tg2 in fibroblast activation (*Figure 4h–i*).

Spheroid cocultures of MDA$^{- (scr)}$ and MDA$^{- (shTg2)}$ with 3T3 fibroblasts were utilized to assess the effects of MV$^{- (shTg2)}$ on cell migration. MDA$^{- (scr)}$ + 3 T3 spheroids cultured with MV$^{- (scr)}$ exhibited a significant increase in spheroid outgrowth compared to untreated conditions, consistent with our prior experiments (*Figure 4j–k*, *Figure 4—figure supplement 2g*). MDA$^{- (shTg2)}$ + 3 T3 spheroids cultured with MV$^{- (shTg2)}$ exhibited no change in spheroid outgrowth compared to untreated control conditions (*Figure 4j–k*, *Figure 4—figure supplement 2g*), suggesting that without Tg2, fibroblasts could not increase cancer cell migration. MDA$^{- (shTg2)}$ +3 T3 spheroids cultured with MV$^{- (scr)}$ exhibited a significant increase in spheroid migration compared to untreated conditions, highlighting that MV-Tg2 is required for MV-mediated fibroblast-induced cancer cell migration (*Figure 4j–k*, *Figure 4—figure supplement 2g*).

To further investigate the role of Tg2 in MV-mediated fibroblast activation, Tg2 was overexpressed in MDA$^+$ using lentiviral transduction (MDA$^{+ (FUW-Tg2)}$) (*Figure 4—figure supplement 3a*). Fibroblasts cultured with MV$^{+ (FUW-Tg2)}$ displayed increased cell area and increased traction force, compared to culture with MV$^+$ (*Figure 4—figure supplement 3b–c*). These results were consistent with earlier findings suggesting that MV-Tg2 regulates MV-mediated fibroblast phenotype (*Figure 4*).

Additionally, Tg2 was overexpressed in the weakly migratory, weakly metastatic MCF7 breast cancer cell line (*Figure 4—figure supplement 3d*). Overexpression of Tg2 did not significantly change the motile fraction of MCF7 in 3D collagen gels (*Figure 4—figure supplement 3e*). Fibroblasts cultured with MV$^{MCF7 (FUW-Tg2)}$ displayed significantly increased cell area compared to control or +MV$^{MCF7}$ conditions and exhibited significantly increased traction force compared to control conditions (*Figure 4—figure supplement 3f–g*). These findings confirm that addition of Tg2 to MVs from a weakly migratory breast cancer cell line regulates MV-mediated fibroblast phenotype. Additionally, spheroid cocultures of MCF7 with 3T3 fibroblasts were utilized to assess the effects of MV-Tg2 on cell migration. MCF7 +3 T3 spheroids cultured in control conditions or with MV$^{MCF7}$ exhibited low levels of spheroid outgrowth at 72 hours (*Figure 4—figure supplement 3h–i*). MCF7 +3 T3 spheroids cultured with MV$^{MCF7 (FUW-Tg2)}$ exhibited significantly higher levels of spheroid outgrowth and MCF7 cells migrating away from the spheroid core were observed (*Figure 4—figure supplement 3h–i*, red arrows). These results reveal that MV-mediated fibroblast activation induced by Tg2 overexpression can enhance the dissemination of the MCF7 weakly migratory breast cancer cell line.

Given that knockdown of Tg2 in MV$^-$ reduced MV$^-$-mediated fibroblast activation and that overexpression of Tg2 in MDA$^+$ or MCF7 increased MV-mediated fibroblast activation, we investigated whether purified Tg2 could induce a change in fibroblast phenotype or whether MV packaging of Tg2 was required for Tg2-induced fibroblast activation. Purified Tg2 (1 µg/mL) significantly increased fibroblast cell area and fibroblast traction force compared to control conditions (*Figure 4—figure supplement 3j–k*). The increased fibroblast area and traction force induced by purified Tg2 were comparable to levels induced by MV$^-$ (*Figure 4—figure supplement 3j–k*). These results reveal that Tg2 alone can induce a change in fibroblast phenotype and that Tg2 packaging by MVs is not essential for Tg2-mediated fibroblast activation.

## Tg2 knockdown in MDA$^-$ reduces metastasis

Given that MV-Tg2 activated fibroblasts in vitro and increased cancer cell migration in spheroid cocultures, we investigated whether knockdown of Tg2 in MDA$^-$ is sufficient to reduce MDA$^-$ metastasis

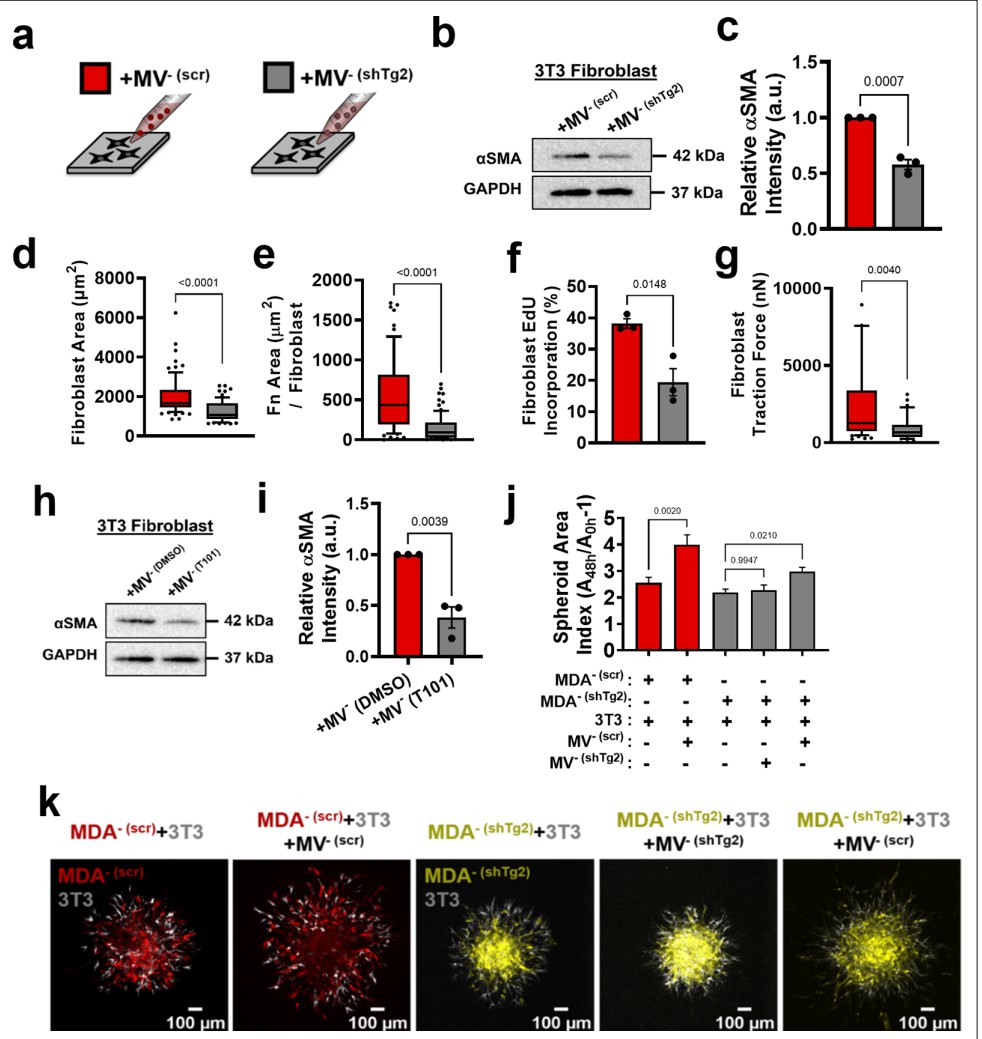

**Figure 4.** Modulation of Tg2 expression in MVs regulates MV-mediated fibroblast activation. (**a**) Schematic overviewing fibroblast culture with MVs. (**b**) Western blot of αSMA and GAPDH in fibroblasts cultured with MV⁻ (scr) (+MV⁻ (scr)) or with MV⁻ (shTg2) (+MV⁻ (shTg2)). (**c**) Quantification of western blot in (**b**) (N=3). (**d**) Fibroblast cell area after culture in + MV⁻ (scr) or + MV⁻ (shTg2) conditions. (N=3; n=75,69). (**e**) Area of fibronectin deposited per fibroblast cultured in + MV⁻ (scr) or + MV⁻ (shTg2) conditions. (N=3; n=62, 82). (**f**) Percentage of fibroblasts EdU positive after culture in + MV⁻ (scr) or + MV⁻ (shTg2) conditions. (N=3). (**g**) Fibroblast traction force after culture in + MV⁻ (scr) or + MV⁻ (shTg2) conditions. (N=3; n=56, 33). (**h**) Western of blot of αSMA and GAPDH in fibroblasts cultured with MV⁻ (DMSO) (+MV⁻ (DMSO)) or with MV⁻ (T101) (+MV⁻ (T101)). (**i**) Quantification of western blot in (**h**) (N=3). (**j**) Spheroid area index quantification 48 hours post embedding. (N=3 + ; n=23, 23, 83, 60, 37). (**k**) Representative images of spheroid outgrowth 48 hours post embedding. MDA⁻ (scr) (red); 3T3 fibroblast (gray); MDA⁻ (shTg2) (yellow). Bar graphs show mean +/- SEM. Box and whisker plots show median and 25th-75th (box) and 10th-90th (whiskers) percentiles. P-values determined using an unpaired Student's t-test or a one-way ANOVA with Tukey's test for multiple comparisons. Source data available in **Source data 1 and 2**.

The online version of this article includes the following figure supplement(s) for figure 4:

**Figure supplement 1.** Modulation of Tg2 expression regulates fibroblast activation cargo in MVs.

**Figure supplement 2.** Modulation of Tg2 expression in MVs regulates MV-mediated fibroblast activation in vitro.

**Figure supplement 3.** Modulation of Tg2 expression in MVs regulates MV-mediated fibroblast activation and spheroid outgrowth in vitro.

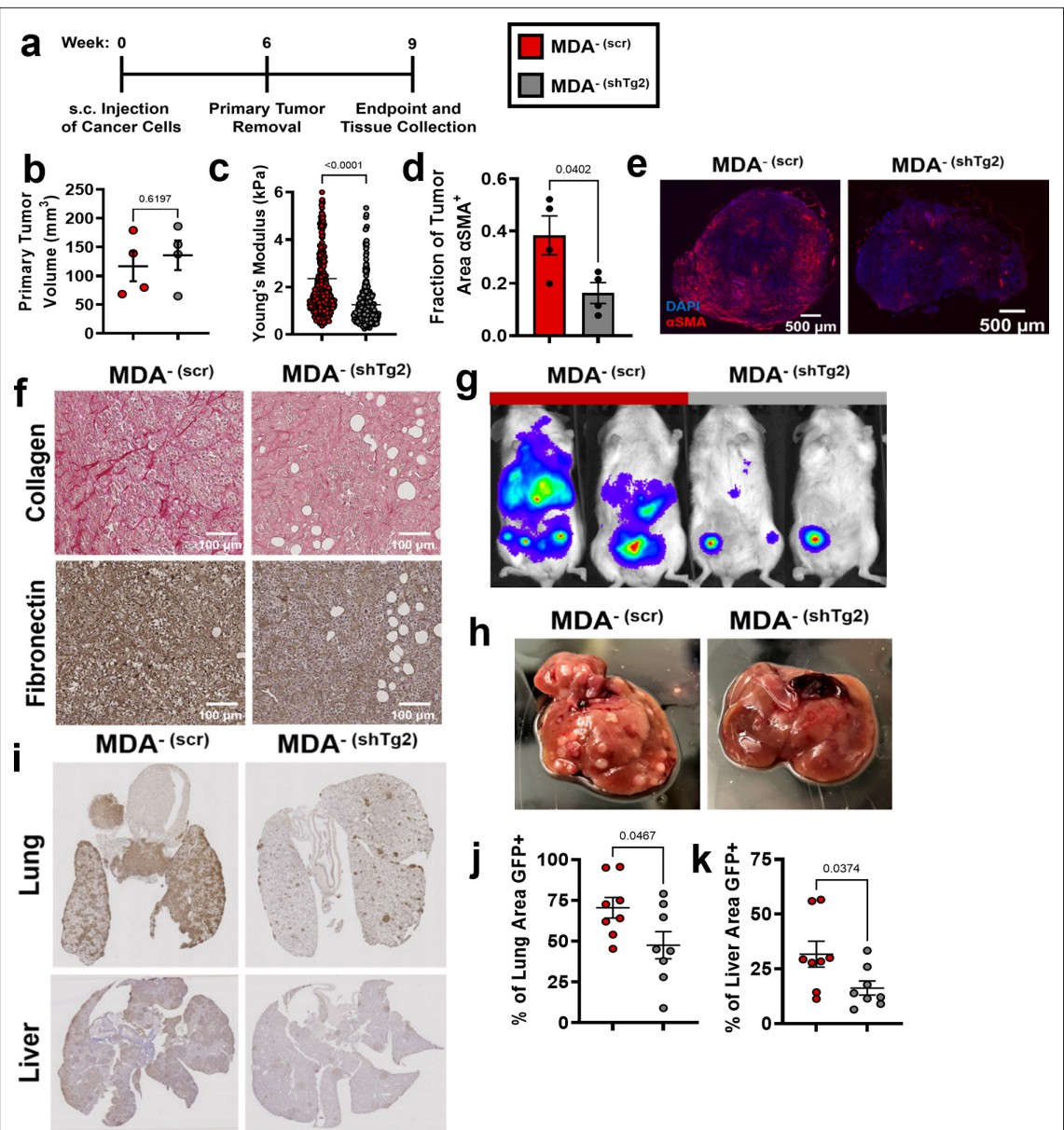

**Figure 5.** Knockdown of Tg2 in MDA⁻ subpopulation reduces metastasis. (**a**) Timeline of orthotopic mammary metastasis model. (**b**) Primary tumor volume after removal (N=4). (**c**) AFM stiffness measurements of MDA⁻ ᐟ(scr) and MDA⁻ ᐟ(shTg2) primary tumors. (N=3 + tumors per condition; n=420, 379). (**d**) Fraction of tumor area positive for αSMA (N=4). (**e**) Representative images of αSMA (red) and DAPI (blue) in MDA⁻ ᐟ(scr) and MDA⁻ ᐟ(shTg2) primary tumors. (**f**) Immunohistochemical staining of collagen and fibronectin in MDA⁻ ᐟ(scr) and MDA⁻ ᐟ(shTg2) primary tumors. (**g**) Representative BLI of MDA⁻ ᐟ(scr) and MDA⁻ ᐟ(shTg2) metastasis. (**h**) Representative image of macroscopic liver nodules. (**i**) Anti-GFP immunohistochemical staining of lungs and liver from MDA⁻ ᐟ(scr) and MDA⁻ ᐟ(shTg2) mice. (**j**) Quantification of percentage of lung tissue area GFP-positive. (N=4, n=8). (**k**) Quantification of percentage of liver tissue area GFP-positive. (N=4, n=8). Data shown as mean +/- SEM. p-Values determined using an unpaired Student's t-test. Source data available in *Source data 2*.

in an orthotopic mammary metastasis mouse model (*Figure 5a*). Six weeks post subcutaneous injection of MDA⁻ ᐟ(scr) and MDA⁻ ᐟ(shTg2) into the mammary fat pad, primary tumors were of similar volumes (*Figure 5b*). MDA⁻ ᐟ(shTg2) tumors exhibited decreased stiffness compared to control tumors (*Figure 5c*). Immunofluorescent staining of MDA⁻ ᐟ(scr) and MDA⁻ ᐟ(shTg2) tumors for the fibroblast activation marker αSMA revealed a decreased fraction of αSMA⁺ tissue in MDA⁻ ᐟ(shTg2) tumors compared to control tumors (*Figure 5d–e*), suggesting that knockdown of Tg2 reduced primary tumor fibroblast activation. Immunohistochemical staining for collagen and fibronectin also revealed decreased matrix in the MDA⁻ ᐟ(shTg2) tumors compared to control tumors (*Figure 5f*). Three weeks post primary tumor removal, BLI imaging and tissue collection was completed. The metastasis of MDA⁻ᐟ(shTg2) to the lungs and liver

was greatly reduced compared to MDA$^{- (scr)}$ (***Figure 5g–k***). Livers from MDA$^{- (shTg2)}$ mice displayed fewer macroscopic liver nodules compared to MDA$^{- (scr)}$ (***Figure 5f***), and immunohistochemical staining for GFP$^+$ cancer cells in mouse liver and lungs revealed decreased GFP + tissue area in the lungs and liver of MDA$^{- (shTg2)}$ mice (***Figure 5g–k***). These findings reveal that knockdown of Tg2 in MDA$^-$ decreases tumor stiffness, fibroblast activation, ECM deposition, and metastasis.

## Tg2-rich MV$^-$ are sufficient to induce MDA$^{- (shTg2)}$ metastasis

Since knockdown of Tg2 in MDA$^-$ reduced MDA$^-$ metastasis, we investigated whether supplementing primary tumors composed of MDA$^{- (shTg2)}$ cells with Tg2-rich MV$^-$ could induce the metastasis of MDA$^{- (shTg2)}$. One week post-subcutaneous injection of MDA$^{- (shTg2)}$ into the mammary fat pad, mice were subcutaneously injected every three days for five weeks with either MV$^-$ suspended in serum-free (SF) media (MDA$^{- (shTg2)}$ + MV$^-$) or an SF media control (MDA$^{- (shTg2)}$ + SF) (***Figure 6a***). After six weeks of primary tumor growth, MDA$^{- (shTg2)}$ primary tumors supplemented with MV$^-$ were significantly larger than and exhibited increased primary tumor stiffness compared to control tumors (***Figure 6b–c***). MDA$^{- (shTg2)}$ + MV$^-$ tumors also exhibited an increased fraction of αSMA$^+$ tissue area compared to control tumors (***Figure 6d–e***). Immunohistochemical staining of tumors revealed increased collagen and fibronectin deposition in the MDA$^{- (shTg2)}$ + MV$^-$ primary tumors compared to control tumors (***Figure 6f***). Three weeks post primary tumor removal, BLI imaging and tissue collection was completed. Supplementing MDA$^{- (shTg2)}$ primary tumors with MV$^-$ induced increased metastasis to both the lungs and liver of mice, compared to control mice (***Figure 6g–k***). Livers from MDA$^{- (shTg2)}$ + MV mice displayed increased numbers of macroscopic liver nodules compared to control mice (***Figure 6h***). Immunohistochemical staining for GFP in mouse lungs revealed increased metastasis to the lungs of MDA$^{- (shTg2)}$ + MV mice compared to control mice (***Figure 6i–j***). Livers of MDA$^{- (shTg2)}$ + MV mice had slightly increased but not significantly higher levels of metastasis compared to control mice (***Figure 6i and k***). These results indicate that repeated injection of Tg2-rich wildtype MV$^-$ during the growth of MDA$^{- (shTg2)}$ primary tumors resulted in primary tumor stiffening, increased fibroblast activation, and increased metastasis of the Tg2-knockdown weakly migratory breast cancer subpopulation.

These findings were additionally confirmed via analysis of MCF7 primary tumors with injections of either serum-free (SF) media or MV$^{MCF7 (FUW-Tg2)}$ suspended in SF media (***Figure 6—figure supplement 1a***). After 8 weeks of primary tumor growth, tumor volumes between MCF7 + SF and MCF7 + MV$^{MCF7 (FUW-Tg2)}$ were not statistically different (***Figure 6—figure supplement 1b***). However, the two tumors of largest volume were from mice that received MV injections. While no metastasis was evident 18 weeks after inoculation in either SF or MV conditions (***Figure 6—figure supplement 1d***), mice that received MV injections had higher numbers of GFP+ cells identified outside of the tumor periphery (***Figure 6—figure supplement 1c***). These results highlight that Tg2-rich MV injection was sufficient to increase cancer cell dissemination into the tumor microenvironment. Importantly, these results also reveal that Tg2 alone is not responsible for inducing metastatic capability in weakly migratory cancer cells. Altogether, our results reveal a novel mechanism by which weakly migratory cancer cells release Tg2-rich MVs to activate fibroblasts and remodel the primary tumor to facilitate weakly migratory cancer cell migration and metastasis (***Figure 6l***).

## Clinical implications of Tg2 expression on breast cancer progression

Given that MV-Tg2 facilitate weakly migratory cancer cell metastasis, we investigated the clinical ramifications of these findings by examining Tg2 expression as a function of patient prognosis. Using the TNMplot database, we found that *TGM2* expression of breast cancer patients increased from normal tissue to breast tumor tissue to metastatic tissue (***Bartha and Győrffy, 2021***), indicating that *TGM2* expression was correlated with breast cancer metastasis (***Figure 7a***). Additionally, breast cancer patients exhibited significantly decreased distant metastasis-free survival with high *TGM2* expression compared to low *TGM2* expression (***Győrffy, 2021***; ***Figure 7b***). This relationship between *TGM2* expression and distant metastasis-free survival was even more robust in triple negative breast cancer (TNBC) patients (***Győrffy, 2021***; ***Figure 7c***). Importantly, *COL12A1*, *PLOD1*, and *PLOD3*, also enriched in MV$^-$ compared to MV$^+$ in ***Figure 3***, did not significantly affect distant metastasis-free survival in TNBC patients (***Figure 7—figure supplement 1a–c***). Lastly, using *TGM2* co-expression data of breast invasive carcinoma patients, CAF markers *ACTA2*, *FAP*, *PDGFRA*, and *PDGFRB* were significantly correlated with *TGM2* expression, indicating that Tg2 expression correlates with fibroblast activation

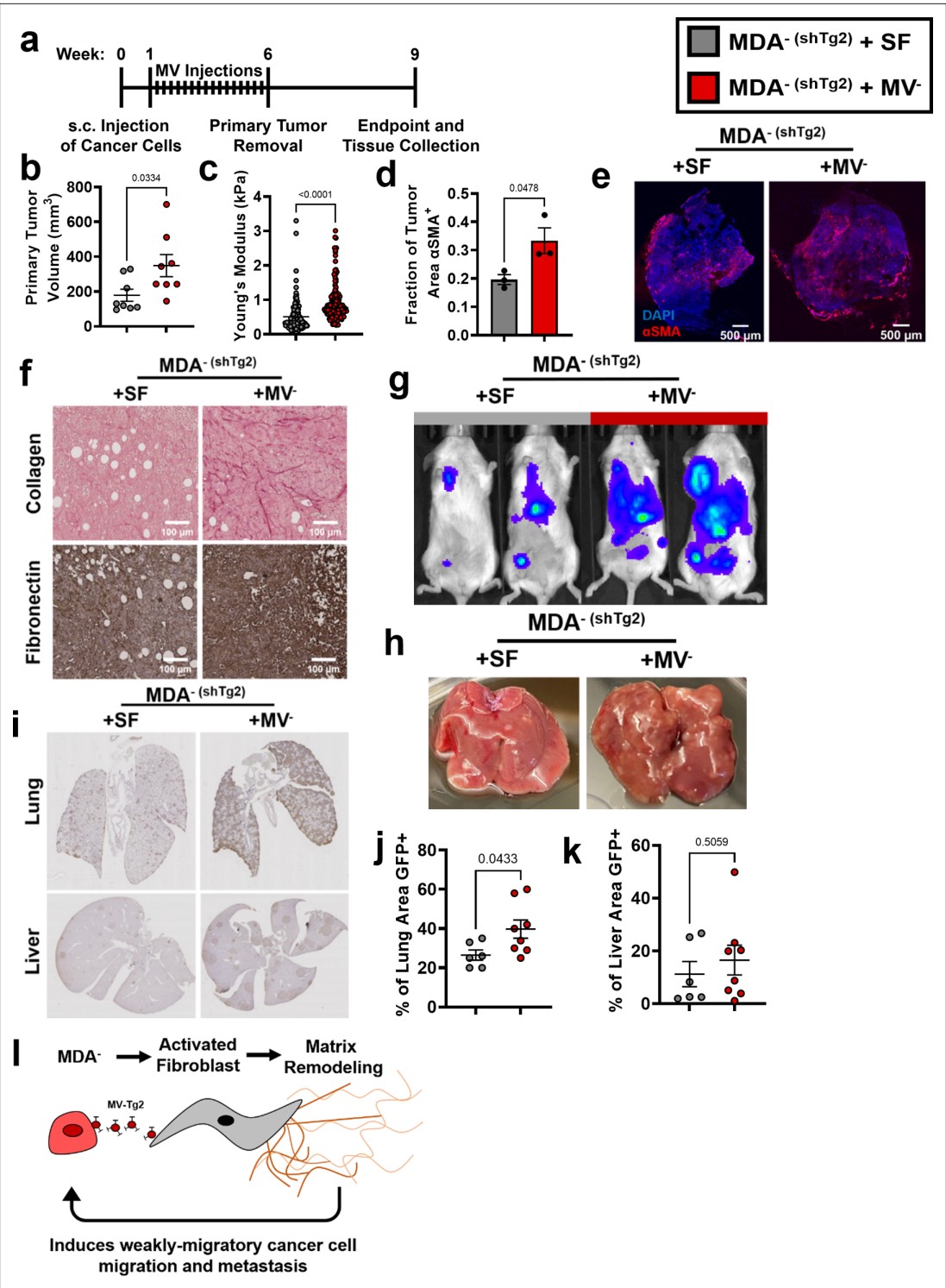

**Figure 6.** Tg2-rich wildtype MV⁻ are sufficient to induce the metastasis of MDA⁻ ⁽ˢʰᵀᵍ²⁾. (**a**) Timeline of orthotopic mammary metastasis model with MV injections every 3 days. (**b**) Primary tumor volume after removal (N=8). (**c**) AFM stiffness measurements of MDA⁻ ⁽ˢʰᵀᵍ²⁾ + SF and MDA⁻ ⁽ˢʰᵀᵍ²⁾ + MV⁻ primary tumors. (N=3 + tumors per condition; n=128, 140). (**d**) Fraction of tumor area positive for αSMA (N=3). (**e**) Representative images of αSMA (red) and DAPI (blue) in MDA⁻ ⁽ˢʰᵀᵍ²⁾ + SF and MDA⁻ ⁽ˢʰᵀᵍ²⁾ + MV⁻ primary tumors. (**f**) Immunohistochemical staining of collagen and fibronectin in MDA⁻ ⁽ˢʰᵀᵍ²⁾ + SF and MDA⁻ ⁽ˢʰᵀᵍ²⁾ + MV⁻ primary tumors. (**g**) Representative BLI of MDA⁻ ⁽ˢʰᵀᵍ²⁾ metastasis. (**h**) Representative image of macroscopic liver nodules. (**i**) Anti-GFP immunohistochemical staining of lungs and liver from MDA⁻ ⁽ˢʰᵀᵍ²⁾ + SF and MDA⁻ ⁽ˢʰᵀᵍ²⁾ + MV mice. (**j**) Quantification of percentage of lung tissue area GFP-positive. (N=6, 8). (**k**) Quantification of percentage of liver tissue area GFP-positive. (N=6, 8). (**l**) Illustration overviewing the mechanism of MDA⁻ metastasis. Data shown as mean +/- SEM. p-Values determined using an unpaired Student's t-test. Source data available in **Source data 2**.

*Figure 6 continued on next page*

*Figure 6 continued*
The online version of this article includes the following figure supplement(s) for figure 6:

**Figure supplement 1.** Primary tumor MV-Tg2 promotes dissemination of weakly migratory cancer cells in vivo.

in breast cancer patients (*Cerami et al., 2012*; *Gao et al., 2013*; *Figure 7d*). Together, these findings further implicate Tg2 as an indicator of breast cancer progression and identify Tg2 as an important therapeutic target to prevent fibroblast activation and metastasis.

## Discussion

Identifying the mechanisms of cancer cell dissemination away from the primary tumor is essential to develop new therapies to target metastatic cancer cells. Specifically, we show that in breast cancer, while highly migratory cells are capable of independent migration, weakly migratory cells release

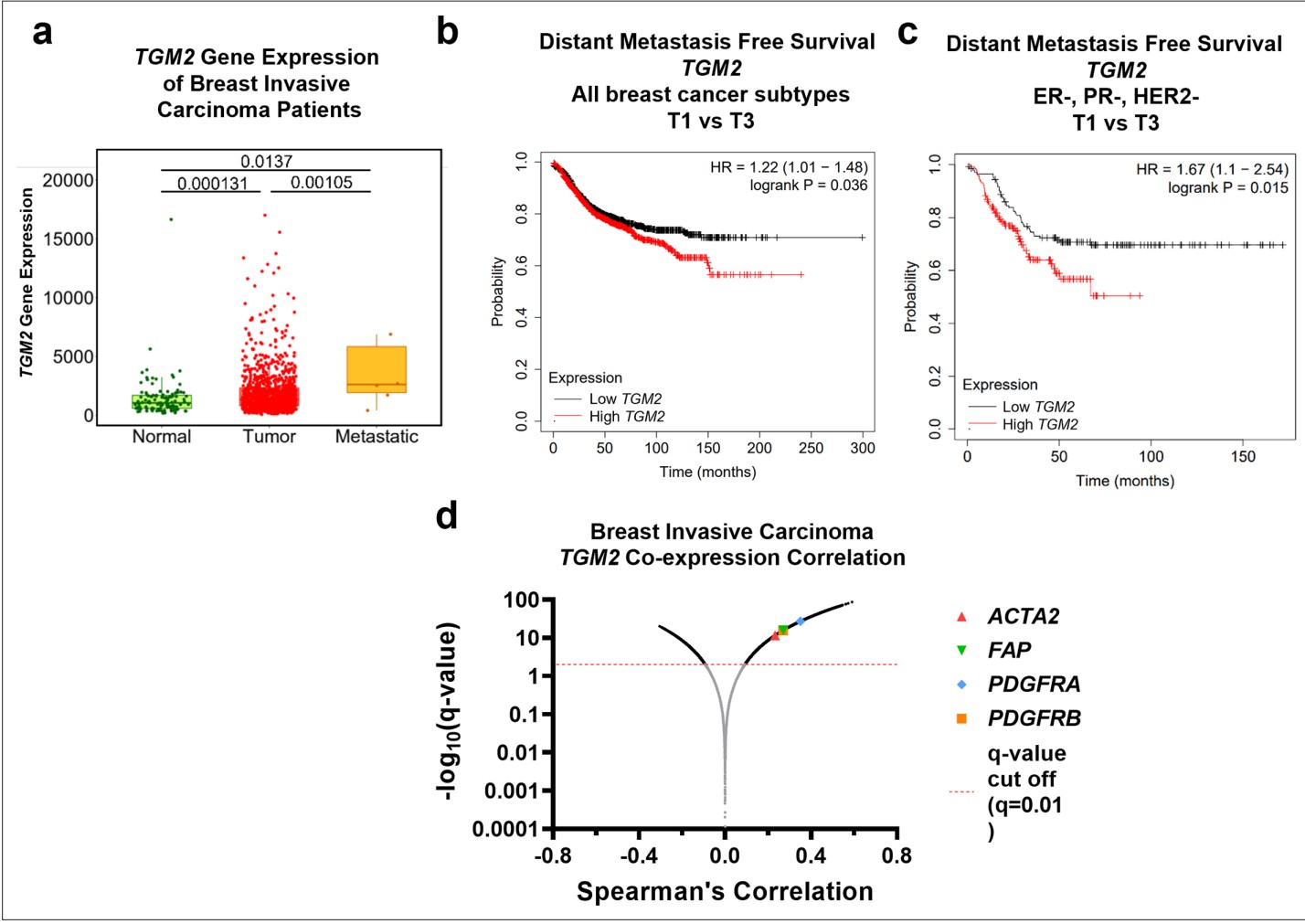

**Figure 7.** Clinical implications of Tg2 expression on breast cancer progression. (**a**) *TGM2* gene expression, measured with RNA sequencing, of normal, tumor, and metastatic tissue from breast invasive carcinoma patients. (N=242, 7569, 82). Data adapted from TNMplot database. (**b**) Distant metastasis-free survival Kaplan-Meier plot for *TGM2* expression in all breast cancer subtypes. Data adapted from the Kaplan-Meier Plotter database. (**c**) Distant metastasis-free survival Kaplan-Meier plot for *TGM2* expression in ER, PR, and HER2 negative breast cancer. Data adapted from the Kaplan-Meier Plotter database. (**d**) *TGM2* co-expression correlation data for breast invasive carcinoma. Data adapted from cBioPortal using the Firehose Legacy dataset. Source data available in *Source data 2*.

The online version of this article includes the following figure supplement(s) for figure 7:

**Figure supplement 1.** COL12A1, PLOD1, and PLOD3 expression do not significantly correlate with distant metastasis free survival in triple negative breast cancer.

Tg2-rich MVs which activate fibroblasts which subsequently promote cancer cell dissemination and ultimately metastasis. Our data suggests that the interaction between weakly migratory cancer cells and stromal fibroblasts via MVs facilitates weakly migratory cancer cell dissemination and metastasis.

Our data supports the observation that cells do not need to be inherently migratory to metastasize (*Hapach et al., 2021*; *Chen et al., 2017*; *Fietz et al., 2017*; *Johnstone et al., 2018*; *Padmanaban et al., 2019*; *Kubens and Zänker, 1998*). Others have shown that MDA-MB-231 isolated from bone metastases were less migratory than their primary tumor counterparts (*Chen et al., 2017*) and that highly metastatic subpopulations of MDA-MB-231 selected through repeated in vivo isolation of metastases were weakly migratory (*Fietz et al., 2017*; *Johnstone et al., 2018*). Our work is the first to describe a mechanism by which cancer cells with a weak migration phenotype can manipulate cells within the tumor microenvironment to escape the primary tumor. Given the current emphasis on cell migration in cancer metastasis research, these findings highlight that cancer cell migration alone is not a sufficient indicator of metastatic potential. A focus on MV-mediated cell-cell signaling between cancer and stromal cells is crucial to fully understand the dynamics of cancer cell dissemination.

Our findings suggest a feedback loop between MV signaling, matrix remodeling, and fibroblast activation may exist. We previously reported that MV-mediated fibroblast activation was enhanced in high stiffness matrices mimicking the breast tumor periphery (*Schwager et al., 2019*). Here, we show that Tg2-rich MVs stiffen the primary tumor, induce matrix deposition by stromal cells, activate fibroblasts, and promote cancer metastasis. Together, these data indicate that MV-induced fibroblast activation leads to increased tumor stiffening through matrix remodeling and this increased stiffness may feedback to prime fibroblasts for continued activation to further enhance cancer metastasis. Additionally, previous work revealed that transformation of normal fibroblasts in vivo by MDA-MB-231 MVs was dependent upon Tg2 (*Schwager et al., 2019*). Our results point to a role for these MV-Tg2-activated fibroblasts in mediating cancer cell escape from the primary tumor through matrix remodeling. As knockdown of Tg2 in MDA-MB-231 is known to decrease metastasis (*Oh et al., 2011*), we suspect this may be due, in part, to reduced MV-mediated fibroblast activation and decreased cancer cell escape from the primary tumor.

Importantly, our work is the first to show that MV cargo varies based on cell phenotype. While it is known cancer cell phenotype affects the numbers of MVs released (*Sedgwick et al., 2015*), our work reveals that differences in cancer cell phenotype within a single cell line actually correlates with differences in MV cargo. Additionally, while previous work showed that highly migratory cancer cells package migration-promoting cargo into EVs (*Steenbeek et al., 2018*), our results suggest that heterogeneity in MV cargo based on migratory phenotype has downstream effects on fibroblast activation, tumor microenvironment remodeling, and ultimately metastasis. Our findings specifically show that Tg2 is not uniformly expressed by all breast cancer-derived MVs but rather dependent upon the migratory phenotype of the MV releasing cell, with increased Tg2 expression in MVs released by the weakly migratory breast cancer cells. Interestingly, the weakly migratory cells express high levels of Tg2 and E-cadherin (*Hapach et al., 2021*). While Tg2 expression is generally linked with EMT and positively correlated with cancer cell invasion and migration (*Lee et al., 2018*; *Wang and Griffin, 2013*; *Mangala et al., 2005*) and E-cadherin is a marker of epithelial phenotypes and decreased cancer cell migration (*Hapach et al., 2021*; *Padmanaban et al., 2019*; *Loh et al., 2019*), Tg2 was recently identified as a marker of epithelial-mesenchymal plasticity and was found to be upregulated in cancer cells undergoing EMT only after a reversion to a secondary epithelial state (*Shinde et al., 2020*). Taken together, these findings reveal that MV cargo changes as cells plastically change their EMT phenotype during the metastasis.

Additionally, our findings suggest that not all MVs released from a cell population will homogenously communicate with surrounding cells. In cancer, targeting MV populations that contain cargo (such as Tg2) to remodel the tumor microenvironment may be crucial to reduce metastatic potential. Importantly, other groups have shown that exosomes affect later stages of cancer metastasis, including angiogenesis and pre-metastatic niche formation (*Shinde et al., 2020*; *Zeng et al., 2018*; *Fang et al., 2018*). As exosomes and MVs contain many of the same cancer-promoting signaling proteins, such as Tg2, VEGF, and TGFβ (*Shinde et al., 2020*; *Feng et al., 2017*; *Ringuette Goulet et al., 2018*; *Ko et al., 2019*; *Yang et al., 2020*), MVs may also signal to endothelial cells to promote angiogenesis and travel to secondary sites to promote pre-metastatic niche formation. Taken together, these findings reveal that identifying the MV populations and MV cargo capable of promoting progression through

the metastatic cascade is essential to prevent metastasis and highlight MVs as a potential target for new anti-metastatic cancer therapies.

In summary, the relationship between cancer cell migratory ability and dissemination away from the primary tumor is complex. While highly migratory cells are capable of independent migration, our work identifies a population of weakly migratory, highly metastatic breast cancer cells which escape the primary tumor via MV-Tg2-mediated fibroblast activation. As we further define the relationship between cancer cell migration and metastatic potential and the consequences of MV signaling on cancer progression, these findings are likely to have broader implications in designing modalities and therapies to detect and target metastatic cancer cells.

## Materials and methods
### Cell culture and reagents
MDA-MB-231 malignant mammary adenocarcinoma cells (HTB-26, ATCC, Manassas, VA), NIH 3T3 fibroblasts, and all modified cell lines were maintained in Dulbecco's Modified Eagle's Media (DMEM) (ThermoFisher Scientific, Waltham, PA) supplemented with 10% fetal bovine serum (FBS) (Atlanta Biologicals, Flowery Branch, GA) and 1% penicillin-streptomycin (ThermoFisher Scientific). MCF7 mammary adenocarcinoma cells (HTB-22; ATCC) were maintained in minimum essential medium (MEM) (ThermoFisher Scientific) supplemented with 10% FBS and 1% penicillin-streptomycin. All cells were cultured at 37 °C and 5% $CO_2$.

Primary antibodies used were rabbit anti-flotillin-2 (3436; Cell Signaling Technology, Danvers, MA), mouse anti-α smooth muscle actin (M0851, DAKO, Santa Clara, CA), mouse anti-beta actin (A5316, Millipore Sigma, Burlington, MA), mouse anti-tissue transglutaminase 2 (ab2386, Abcam, Cambridge, UK), goat anti-fibronectin (sc6953, Santa Cruz, Dallas, TX), rabbit anti-fibronectin (F3648, Millipore Sigma), rabbit anti-focal adhesion kinase (3285, Cell Signaling Technology, Danvers, MA), rabbit anti-p-focal adhesion kinase (Tyr397) (3283, Cell Signaling Technology), and mouse-anti GAPDH (MAB374, Millipore Sigma). Secondary antibodies used were HRP anti-rabbit (Rockland, Limerick, PA), HRP anti-mouse (Rockland), AlexaFluor 488 conjugated to donkey anti-goat (Life Technologies, Carlsbad, CA), AlexaFluor 488 conjugated to donkey anti-rabbit (Life Technologies), AlexaFluor 488 conjugated to donkey anti-mouse (Life Technologies), and AlexaFluor 568 conjugated to donkey anti-mouse (Life Technologies). Actin was stained using Texas Red Phalloidin (Life Technologies).

### Phenotypic sorting of MDA-MB-231 breast cancer cells
MDA-MB-231 breast cancer cells were phenotypically sorted based on their ability to migrate through a collagen gel on top of a Transwell insert as previously described (*Hapach et al., 2021*). Briefly, cancer cells were seeded on a thin layer of 1 mg/ml collagen (Corning, Corning, NY) on top of a Transwell insert with 8 µm pores (Greiner Bio-One, Kremsmunster, Austria). A serum gradient was applied and cancer cells were allowed to migrate for four days. After four days, highly migratory and weakly migratory cells were collected and reseeded in fresh Transwells. After 20 rounds of purification, cells that repeatedly migrated through the assay were termed 'highly migratory' (MDA⁺) and cells that never migrated through the assay were termed 'weakly migratory' (MDA⁻).

### Modified cell lines
MDA⁺ and MDA⁻ were stably transduced with either FUW-GFP-E2A-fluc or FUW-mCherry-E2A-rluc, both created in-house. NIH 3T3 fibroblasts were transduced with either Life-Act eGFP (#84383, Addgene, Watertown, MA) or FUW-mCherry-E2A-rluc. To generate a Tg2-knockdown and a scrambled control cell line, MDA⁻ ⁽ᴳᶠᴾ/ˡᵘᶜ⁾ were subjected to lentiviral transduction with either the Tg2-targeting MISSION shRNA plasmid (SHCLND-NM_004613: TRCN0000272816) (MDA⁻ ⁽ˢᶜʳ⁾) or the MISSION scr.1-puro scrambled control plasmid DNA (SHC001; Millipore Sigma) (MDA⁻ ⁽ˢʰᵀᵍ²⁾). To generate Tg2 over-expressing cell lines, MDA⁺ and MCF7 were stably transduced with an FUW-Tg2 plasmid created in-house.

### Orthotopic mammary metastasis mouse model
Orthoptic injection of NOD SCID gamma (NSG) mice was conducted as previously described (*Hapach et al., 2021*). Briefly, female NSG mice, 6–8 weeks of age, were injected subcutaneously at the fourth

mammary fat pad with cancer cells lentivirally transduced with GFP and firefly luciferase tags. In mouse experiments with MVs, either serum-free media alone or approximately $1*10^7$ MV$^-$ suspended in serum-free media were injected subcutaneously at the primary tumor site every three days from one to six weeks (MDA$^-$ study) or eight weeks (MCF7 study) post cancer cell injection. For bioluminescent imaging, mice were injected with 30 mg/mL D-luciferin (Gold-Bio, St Louis, MO) and imaged weekly on an IVIS Lumina III Series (Caliper LifeSciences, Hopkinton, MA). After 4–10 weeks, primary tumors were removed using sterile surgical technique. Primary tumor size was measured with calipers. Primary tumors were cut in half and either snap-frozen in dry ice or fixed with 4% v/v paraformaldehyde (Electron Microscopy Sciences, Hartfield, PA). At the endpoint, mouse lung and liver were collected and fixed in 4% v/v paraformaldehyde for 24 hr and sent to the Vanderbilt Tissue Pathology Shared Resource for paraffin embedding and sectioning.

## Atomic force microscopy of primary mouse tumors

Primary tumor samples cryopreserved in O.C.T compound were cut into 20 µm sections. Prior to AFM stiffness measurements, samples were thawed at room temperature for 3 min, and maintained in 1 X Halt Protease Inhibitor Cocktail (78438, ThermoFisher Scientific). A thermoplastic coverslip (ThermoFisher Scientific) with a 6 µm biopsy punch hole was superglued on top of the tumor section. An ImmEdge hydrophobic pen (Vector Laboratories, Burlingame, CA) was used to draw a small circle around the biopsy punch hole. AFM measurements were performed using a MFP3D-BIO inverted optical AFM (Asylum Research, Santa Barbara, CA), with an inverted fluorescent Zeiss Observer Z.1 microscope with a 10 x/0.3 N.A. objective. A silicon nitride cantilever with a 5 µm diameter spherical borosilicate glass tip and a spring constant of 0.06 N/m (Novascan Technology, Boone, IA) was used. Samples were indented at a 2 µm/s loading rate, with a maximum indentation force of 5 nN. To obtain tumor stiffness measurements, IGOR PRO Software (Asylum Research) was used. At least two force maps of each primary tumor sample were obtained. AFM data was fit to a Hertz model with a Poisson's ratio of 0.5.

## qPCR of tumors

Snap frozen primary tumors were prepared for RNA isolation by TRIZOL digestion and subsequent homogenization using a TissueLyser II (Qiagen, Hilden, Germany) with a 5 mm stainless steel bead for 2 min at 30 Hz. After homogenization, samples were incubated at room temperature for 5 min in TRIZOL. Chloroform was added to the homogenized sample and incubated at room temperature for 3 min. Samples were centrifuged at 10,000 x g for 30 min at 4 °C. The upper aqueous phase was separated and mixed with 70% ethanol. RNA was subsequently isolated using the RNeasy Mini Kit (Qiagen). DNA was synthesized from RNA using the iScript cDNA Synthesis Kit (Bio-Rad Laboratories, Hercules, CA). qPCR was performed with the iQ SYBR Green Supermix (Bio-Rad Laboratories) per manufacturer protocols. The primer sequences for each gene are listed in *Supplementary file 1*.

## Immunohistochemical staining of mouse tissues

Primary tumors were fixed, paraffin-embedded, and mounted into 5-µm-thick sections at the Vanderbilt Tissue Pathology Shared Resource. For collagen staining, tumors were stained using the picrosirius red stain kit (Polysciences Inc, Warrington, PA). For fibronectin staining, tumors were stained using the Abcam IHC-Paraffin Protocol. Briefly, 5-µm-thick sections of primary tumors were deparaffinized and rehydrated using a series of washes of xylene, ethanol, and water. Heat-induced epitope retrieval was completed using a sodium citrate buffer. Tumor sections were blocked for 2 hr in tris buffered saline (TBS) + 10% fetal bovine serum +1% bovine serum albumin at room temperature. Samples were incubated overnight in rabbit anti-fibronectin (1:500) in TBS + 1% bovine serum albumin. After washing, samples were incubated in TBS + 0.3% hydrogen peroxide for 15 min to suppress endogenous peroxidase activity. Samples were incubated in HRP anti-rabbit (1:200) for 1 hr. Staining was developed using the DAB chromogen (8059 S, Cell Signaling Technology) and samples were subsequently counterstained using Mayer's hematoxylin (Millipore Sigma). Tumor samples were then dehydrated, cleared, and mounted. Anti-GFP staining was completed by the Vanderbilt Tissue Pathology Shared Resource. Using brightfield (3,3'-diaminobenzidine) image analysis in the open-source software QuPath, digitized images of the primary tumors were annotated to define the tumor edge and surrounding tissue. Default parameters were used for positive cell detection with a 3 µm

cell expansion parameter. The GFP Positive Cells per Area value was calculated by dividing the total number of detected positive cells by the area of the peripheral tissue surrounding the primary tumor (in $mm^2$). Whole slide imaging and quantification of anti-GFP immunostaining of liver and lungs were performed in the Digital Histology Shared Resource at Vanderbilt University Medical Center (http://www.mc.vanderbilt.edu/dhsr).

## Immunofluorescent staining of mouse tissues
Primary tumors were frozen and cut into 20-μm-thick sections at the Vanderbilt Tissue Pathology Shared Resource. Tumor sections were thawed, fixed for 10 min in 4% v/v paraformaldehyde, and permeabilized in 1% Triton X-100 (Millipore Sigma). Sections were blocked for 2 hr in PBS + 10% fetal bovine serum + 5% donkey serum + 5% goat serum. Samples were incubated overnight in mouse anti-αSMA (1:200). After washing, samples were incubated in AlexaFluor 568 conjugated to donkey anti-mouse (1:200) and DAPI (1:500) for 2 hr. Sections were mounted for imaging using Vectashield Mounting Media (Vector Laboratories). Sections were imaged with a 10 x/0.3 N.A. objective on a Zeiss LSM800 confocal laser-scanning microscope.

## RNA sequencing
RNA was isolated from MDA$^+$ and MDA- cultured on tissue culture plastic using the RNAeasy mini-kit (Qiagen) and RNase-free DNase Set (Qiagen, Hilden, Germany). RNA sequencing was performed by the Vanderbilt Technologies for Advanced Genomics Core as previously described (*Hapach et al., 2021*).

## MV isolation and characterization
MVs were isolated as previously described. MDA-MB-231 and MCF7 were incubated overnight in serum-free media. The conditioned media was removed from the cells and centrifuged at 400 RPM. The supernatant was removed and again centrifuged at 400 RPM. The medium was then filtered through a 0.22 μm SteriFlip filter unit (Millipore Sigma) and rinsed with serum-free media. The MVs retained by the filter were resuspended in serum-free media. Nanoparticle tracking analysis (ZetaView ParticleMetrix, Germany) was used to determine the size and number of isolated MVs.

## Western blotting
Isolated MVs were rinsed with PBS on a 0.22 μm SteriFlip filter unit and lysed with Laemmli buffer. MDA$^+$, MDA$^-$, and NIH 3T3 fibroblasts were lysed with Laemmli buffer. For T101 experiments, MV$^-$ were incubated for 30 min with either DMSO or 10 μM T101. MVs were re-filtered and resuspended in serum-free media. NIH 3T3 fibroblasts were cultured with either MV$^{- (DMSO)}$ or MV$^{- (T101)}$ for 24 hr before lysis with Laemmli buffer. All lysates were resolved by SDS-PAGE and transferred to PVDF membranes. Transferred membranes were blocked with either 5% milk or 5% bovine serum albumin in TBS-Tween. Membranes were incubated overnight in mouse anti-α smooth muscle actin (1:1000), rabbit anti-flotillin-2 (1:1000), mouse anti-tissue transglutaminase 2 (1:1000), rabbit anti-focal adhesion kinase (1:1000), rabbit anti-phospho focal adhesion kinase (1:1000), and mouse anti-GAPDH (1:2000) at 4 °C. Membranes were then incubated in horseradish peroxidase-conjugated secondary antibody (1:2000) in 5% milk or 5% bovine serum albumin in TBS-Tween for 1 hr at room temperature. Samples were imaged with an Odyssey Fc (LI-COR Biosciences, Lincoln, NE) after the addition of SuperSignal West Pico or West Dura Chemiluminescent Substrates (ThermoFisher Scientific).

## Polyacrylamide gel preparation
Polyacrylamide (PA) gels were fabricated as described elsewhere (*Califano and Reinhart-King, 2008*). Briefly, the ratio of acrylamide (40% w/v; Bio-Rad, Hercules, CA) to bis-acrylamide (2% w/v; Bio-Rad) was varied to tune gel stiffness to 20 kPa, previously shown to enhance fibroblast response to MVs (*Schwager et al., 2019*). 20 kPa PA gels were fabricated using an acrylamide: bis-acrylamide ratio of 12%:0.19%. PA gels were coated with 0.1 mg/ml rat tail type I collagen (Corning).

## Fibronectin immunofluorescence and analysis
NIH 3T3 fibroblasts were seeded on 20 kPa PA gels in 1.6 mL of DMEM + 1% FBS. Cells were treated with either 400 μL of serum-free media or approximately $3*10^7$ MVs suspended in 400 μL serum-free

media for 24 hr. Cells were first fixed with 3.2% v/v paraformaldehyde and subsequently permeabilized with a 3:7 ratio of methanol to acetone. Cells were incubated with goat anti-fibronectin (1:100) overnight. The cells were washed and incubated with AlexFluor 488 conjugated to donkey anti-goat (1:200), TexasRed phalloidin (1:500), and DAPI (1:500) for 1 hr. To image, gels were inverted onto a drop of Vectashield Mounting Media placed on a glass slide. Fluorescent images were acquired with a 20 x/1.0 N.A. water-immersion objective on a Zeiss LSM700 Upright laser-scanning microscope. Cells stained with phalloidin were outlined in ImageJ to calculate cell area. Area of fibronectin was measured by using the threshold function in ImageJ to calculate area of positive stain. The ratio of fibronectin area to cell area was calculated.

## Phalloidin and αSMA immunofluorescence and analysis

NIH 3T3 fibroblasts were seeded on 20 kPa PA gels in 1.6 mL of DMEM +1% FBS. Cell media was additionally supplemented with either 400 µL of serum-free media or approximately $3*10^7$ MVs suspended in 400 µL serum-free media. After 24 hr, cells were fixed with 3.2% v/v paraformaldehyde and permeabilized with 1% Triton X-100. Cells were blocked with 3% bovine serum albumin in 0.02% Tween in PBS and then incubated for 3 hr at room temperature with mouse anti-alpha smooth muscle actin (1:100). After being washed, cells were incubated for 1 hr with AlexaFluor 488 conjugated to donkey anti-mouse (1:200). The cells were washed and F-actin and nuclei were stained with TexasRed phalloidin (1:500) and DAPI (1:500), respectively. To image, gels were inverted onto a drop of Vectashield Mounting Media placed on a glass slide. Fluorescent images were acquired with a 20 x/1.0 N.A. water-immersion objective on a Zeiss LSM700 Upright laser-scanning microscope.

## EdU proliferation assay

NIH 3T3 fibroblasts were serum-starved for 6 hr and subsequently seeded on 20 kPa PA gels in 1.6 mL of DMEM + 1% FBS. Cell media was additionally supplemented with either 400 µL of serum-free media or approximately $3*10^7$ MVs suspended in 400 µL serum-free media. After 24 hr, 10 µM 5-ethynyl-2′-dexoyuridine (EdU, ThermoFisher Scientific) was added to the culture media for 2 hr. Cells were fixed with 3.2% v/v paraformaldehyde and stained with the Click-iT EdU Kit (ThermoFisher Scientific) following the manufacturer's instructions. Nuclei were counterstained with DAPI (1:500). Cells were imaged with a 20 x/1.0 N.A. water-immersion objective on a Zeiss LSM700 Upright laser-scanning microscope. The percentage of EdU incorporation was calculated as the ratio of EdU positive cells to the total number of cells.

## Traction force microscopy

Traction force microscopy was performed as previously described (*Kraning-Rush et al., 2012*). Briefly, 20 kPa PA gels, embedded with 0.5 µm diameter fluorescent beads (ThermoFisher Scientific), were prepared. NIH 3T3 fibroblasts were allowed to adhere for 24 hr in 1.6 mL of DMEM +1% FBS supplemented with either 400 µL of serum-free media or approximately $3*10^7$ MVs suspended in 400 µL serum-free media. For recombinant Tg2 studies, cells were cultured with 1 µg/mL human recombinant tissue transglutaminase (ENZ-394, Prospec Bio, Rehovot, Israel). After 24 hr, phase contrast images of fibroblasts and fluorescent images of the beads at the surface of the PA gel were acquired. Fibroblasts were removed from the PA gel using 0.25% trypsin/EDTA (Life Technologies) and a fluorescent image of the beads was acquired after cell removal. Bead displacements between the stressed and null states and fibroblast area were calculated and analyzed using the LIBTRC library developed by M. Dembo (Dept. of Biomedical Engineering, Boston University) (*Dembo and Wang, 1999*). Outliers were removed using the ROUT method with Q=0.2%.

## Spheroids

MDA-MB-231, MCF7, and/or NIH 3T3 fibroblasts, transduced with either GFP or mCherry, were resuspended in spheroid compaction media containing 0.25% methylcellulose (STEMCELL, Vancouver, Canada), 4.5% horse serum (ThermoFisher Scientific), 18 ng/mL EGF (ThermoFisher Scientific), 90 ng/mL cholera toxin (Millipore Sigma), 90 U/mL penicillin (Life Technologies), and 90 µg/mL streptomycin (Life Technologies) in DMEM/F12 (ThermoFisher Scientific). A 2:1 ratio of cancers cells to fibroblasts (1:1 for MCF7:3T3 spheroids) were added to wells of a round-bottom 96-well plate to generate 5500 cell spheroids. The plate was centrifuged at 1100 rpm for 5 min at room temperature and

subsequently incubated at 37 degrees for 72 hr (24 hr for MCF7 +3 T3 spheroids) to allow for spheroid compaction.

After spheroid compaction, spheroids were embedded into 4.5 mg/mL collagen gels. Briefly, 4.5 mg/mL collagen gels were generated by mixing 10 mg/ml stock collagen (Rockland), 0.1% acetic acid, culture media, 10 x HEPES (Millipore Sigma), and 1 N NaOH (Millipore Sigma) and added to wells of a 24 well plate. Compacted spheroids were placed into the middle of the collagen gel without touching the bottom of the well plate. The embedded spheroids were placed in a 37 °C incubator for 10 min to polymerize, then the plate was flipped upside-down for another 30 min to avoid spheroid adhesion on the bottom of the well. Spheroids were rehydrated every 24 hr with 300 μL of 1% DMEM and supplemented with either 200 μL of serum-free DMEM or $1.5*10^7$ MVs suspended in 200 μL serum-free DMEM. Spheroids were imaged every 24 hr for 48 hr (72 hr for MCF7 + 3 T3 spheroids) using a Zeiss LSM800 inverted confocal microscope equipped with an environmental control chamber. Spheroid images were captured using a 10 X/0.3 NA objective. The projected spheroid area, the spheroid diameter, and the maximum cancer cell migration distance from the spheroid core was measured. The expansion index ($A_i/A_0$-1) was calculated to quantify spheroid outgrowth.

## Proteomics and analysis

After MV isolation from $MDA^+$ and $MDA^-$, MVs were lysed in a buffer composed of 2% Nonidet P-40 (Millipore Sigma), 0.5% sodium deoxycholate (Millipore Sigma), 300 mM sodium chloride (Millipore Sigma), and 50 mM Tris pH 8. Lysates were incubated for 30 min at 4 °C and centrifuged at 14,000 x g for 15 min at 4 °C to pellet non-solubilized proteins. iTRAQ proteomics of lysates was completed by the Vanderbilt Mass Spectrometry Research Center Proteomics Core. Briefly, enzyme digestion of lysates was used to generate proteolytic peptides. $MV^+$ and $MV^-$ peptides were labeled with 117 and 115 iTRAQ reagents, respectively. Samples were subsequently mixed, fractioned using liquid chromatography, and analyzed via tandem mass spectrometry. A database search using the fragmentation data identified the labeled peptides and their corresponding proteins.

For comparing the protein expression of MVs from $MDA^{- (scr)}$ and $MDA^{- (shTg2)}$ cells, MVs were isolated and concentrated by ultracentrifugation at 100,000 x g for 1 hr at 4°C. The concentrated MV pellet was lysed in a buffer composed of 1 x Cell Lysis Buffer (Cell Signaling Technologies) and complete EDTA-free protease inhibitor cocktail (Sigma-Aldrich). TMT proteomics of lysates was completed by the Vanderbilt Mass Spectrometry Research Center Proteomics Core. Briefly, enzyme digestion of lysates was used to generate proteolytic peptides. $MV^{- (scr)}$ and $MV^{- (shTg2)}$ peptides were labeled with 130 and 131 TMT reagents, respectively. Samples were subsequently mixed, fractionated using liquid chromatography, and analyzed via tandem mass spectrometry. A database search using the fragmentation data identified the labeled peptides and their corresponding proteins.

Protein set enrichment analysis was completed using the PSEA-Quant algorithm (*Lavallée-Adam et al., 2014*). The REVIGO web tool was used to remove redundant GO terms (*Supek et al., 2011*).

## Statistical analysis

All statistical analysis was performed using GraphPad Prism 7 (GraphPad Software, La Jolla, CA) or Excel 2016 (Microsoft, Redmond, WA). Where appropriate, data were compared with a student's t-test or with a two-way analysis of variance (ANOVA) with Sidak multiple comparisons test. All data is reported as mean ± standard error (SE) unless otherwise notated.

## Acknowledgements

This work was supported by the WM Keck Foundation, the National Institutes of Health (GM13117) to CAR-K. This work was also supported by National Science Foundation Graduate Research Fellowship Awards under Grant No. 1937963 to SCS and JAM and Grant No. DGE-1650441 to LAH and the Scholarship for the Next Generation of Scientists from the Cancer Research Society, the National Cancer Institute Grant K99CA212270, and the Canada Research Chair program awarded to FB We thank Alissa Weaver for the use of the ZetaView ParticleMetrix. We acknowledge the Tissue Pathology Shared Resource supported by NCI/NIH Cancer Center Support Grant 5P30 CA68485-19, the Vanderbilt Mouse Metabolic Phenotyping Center Grant 2 U24 DK059637-16, and the Shared Instrumentation Grant S10 OD023475-01A1 for the Leica Bond Rx.

# Additional information

## Funding

| Funder | Grant reference number | Author |
|---|---|---|
| W. M. Keck Foundation | | Cynthia A Reinhart-King |
| National Institute of General Medical Sciences | GM13117 | Cynthia A Reinhart-King |
| National Science Foundation | 1937963 | Samantha C Schwager Jenna A Mosier |
| National Science Foundation | DGE-1650441 | Lauren A Hapach |
| Cancer Research Society | | Francois Bordeleau |
| National Cancer Institute | K99CA212270 | Francois Bordeleau |
| National Cancer Institute | 5P30 CA68485-19 | Cynthia A Reinhart-King |
| National Institute of Diabetes and Digestive and Kidney Diseases | U24 DK059637-16 | Cynthia A Reinhart-King |

The funders had no role in study design, data collection and interpretation, or the decision to submit the work for publication.

## Author contributions

Samantha C Schwager, Conceptualization, Data curation, Formal analysis, Funding acquisition, Investigation, Visualization, Methodology, Writing – original draft, Writing – review and editing; Katherine M Young, Formal analysis, Investigation, Writing – review and editing; Lauren A Hapach, Francois Bordeleau, Conceptualization, Funding acquisition, Writing – review and editing; Caroline M Carlson, Anissa L Jayathilake, Data curation, Formal analysis; Jenna A Mosier, Data curation, Formal analysis, Funding acquisition; Tanner J McArdle, Formal analysis; Wenjun Wang, Data curation; Curtis Schunk, Madison E Bates, Formal analysis, Investigation; Marc A Antonyak, Richard A Cerione, Conceptualization; Cynthia A Reinhart-King, Conceptualization, Resources, Supervision, Funding acquisition, Investigation, Project administration, Writing – review and editing

## Author ORCIDs

Samantha C Schwager ⓘ http://orcid.org/0000-0001-7927-3156
Wenjun Wang ⓘ http://orcid.org/0000-0003-0907-6282
Francois Bordeleau ⓘ http://orcid.org/0000-0002-5114-1757
Cynthia A Reinhart-King ⓘ http://orcid.org/0000-0001-6959-3914

## Ethics

Experiments were performed in accordance with AAALAC guidelines and were approved by the Vanderbilt University Institutional Animal Care and Use Committee (Protocol No. M1700029-00).

## Decision letter and Author response

Decision letter https://doi.org/10.7554/eLife.74433.sa1
Author response https://doi.org/10.7554/eLife.74433.sa2

# Additional files

## Supplementary files

- Transparent reporting form
- Source data 1. Raw western blot image files.
- Source data 2. Raw numerical data.
- Supplementary file 1. Primer sequences used for quantitative real-time PCR.

## Data availability

Source data is included in supporting files. All supporting data sheets contain the figures in the file name and the figure panel in the excel tab.

The following dataset was generated:

| Author(s) | Year | Dataset title | Dataset URL | Database and Identifier |
|---|---|---|---|---|
| Schwager S, Young K, Schunk C, Bates M, Hapach L, Carlson C, Mosier J, McArdle T, Wang W, Schunk C, Jayathilake A, Bates M, Bordeleau F, Antonyak M, Cerione R, Reinhart-King C | 2022 | Weakly migratory metastatic breast cancer cells activate fibroblasts via microvesicle-Tg2 to facilitate dissemination and metastasis | http://doi.org/10.5061/dryad.4qrfj6qbd | Dryad Digital Repository, 10.5061/dryad.4qrfj6qbd |

The following previously published datasets were used:

| Author(s) | Year | Dataset title | Dataset URL | Database and Identifier |
|---|---|---|---|---|
| Győrffy B | 2021 | TNMplot: differential gene expression analysis in Tumor, Normal, and Metastatic tissues | https://tnmplot.com/analysis/ | Genomic Data Commons, tnmplot |
| Győrffy B | 2021 | Kaplan-Meier Plotter | https://kmplot.com/analysis/ | Semmelweis University, kmplot |

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
