## [Editor Report]

This is an important study demonstrating that poorly migratory breast cancer cells can be metastatic by activating fibroblasts via Tg2-containing microvesicles (MVs). A convincing array of methodologies reveal the metastasis-promoting qualities of MV-associated Tg2 and also demonstrates that cancer cells can communicate with the microenvironment in order to overcome tumour-cell intrinsic deficiencies in migratory capacity. This work will be of interest to cancer biologists studying tumour heterogeneity and the role of the tumour microenvironment in metastatic progression.

---

## [Decision Letter]

**Decision letter after peer review:**

Thank you for submitting your article "Weakly migratory metastatic breast cancer cells activate fibroblasts via microvesicle-Tg2 to facilitate dissemination and metastasis" for consideration by *eLife*. Your article has been reviewed by 2 peer reviewers, including Lynne-Marie Postovit as Reviewing Editor and Reviewer #1, and the evaluation has been overseen by Richard White as the Senior Editor. The following individual involved in review of your submission has agreed to reveal their identity: Dennis Discher (Reviewer #2).

Essential revisions:

1) The study should be extended to determine whether the results are cell-line specific. While focused on the TME, the study uses only a human cell-line derived xenograft model. At the very least, some of the experiments should be repeated in a syngeneic model (To account for immune cells and mouse-specific cytokine signaling) and/or an additional human model (PDX and/or cell line).

2) Mechanistic studies should be enhanced. This could include a more thorough examination of CAF phenotypes or studies that better consider previous results using the same model.

3) The authors should describe factors within MVs that may also mediate the observed effects. Also, the authors should address whether other proteins (including MMP-14) change with Tg-2 knockdown; and if these alterations may mediate the effects of Tg-2 on parameters such as survival. Finally, the effects of pure Tg-2 should be compared to the effects of Tg-2 containing-MVs.

4) Presentation of the proteomics data must be improved. The authors should show a PCA to demonstrate the extent to which the samples differ and to also determine variability in biological replicates. Differences between EV- and EV+ cargo should be displayed in a volcano plot that shows the fold difference of specific proteins as well as the significance of such differences when replicates are considered.

5) The data in 2 D and E do not convincingly demonstrate differential fibroblast activation as loading seems uneven. This work should include gene expression data, replicates, and a time course.

6) A serum supplemented control should be used for the in vivo experiments conducted in Figure 6, to better match the experimental conditions.

7) Details of culture conditions for generation of MV should be provided in the text, and justified as to the in vivo tumor relevance of MV generation.

8) In figure 7, cutoffs should be indicated. Moreover, Tg-2 should be correlated to the transcriptional signature of poorly migratory cells and/or stromal scores in order to validate the notion that it mediates the metastasis of this subset of cells through fibroblast activation.

*Reviewer #1 (Recommendations for the authors):*

The current manuscript by Schwager and colleagues describes a mechanism by which poorly migratory MDA-MB-231 cells can be metastatic. This study follows a recent paper from the same group (published in January) demonstrating that these poorly migratory cells are more metastatic than their highly migratory counterparts, and that this is due at least in part to E-Cadherin expression and the ability to form circulating tumour cell (CTC) clusters. In the current study, the authors show that the low migratory cells secrete unique EVs that can activate fibroblasts, concomitant with metastatic progression, and that this function is dependent on the presence of Tg-2. The novelty of this work is in the phenotypic heterogeneity of tumour cells, even within cell lines, and the importance the microenvironment in mediating metastasis associated with this diversity. While interesting, this work uses only one model, which was very recently published. The study, I think, would require repetition within additional models, as well as the inclusion of mechanistic studies designed to determine why the EV cargo differs between the highly and poorly migratory subclones.

1) Fibroblasts are also heterogeneous. CAF populations in migratory versus poorly migratory tumours should be more thoroughly explored to account for the possibility that CAF subtypes change between conditions and in response to EVs.

2) The current study does not really reconcile the mechanisms revealed in the published paper that uses the same model. For example, it does not attempt to compare RNAseq data with proteomics data or to determine whether EVs affect E-Cadherin expression. It also does not explore whether CAFs may be needed for the formation of CTC clusters.

3) While focused on the TME, the study uses only a human cell-line derived xenograft model. At the very least, the work should be repeated in a syngeneic model (To account for immune cells and mouse-specific cytokine signaling) as well as several more human models, ideally including PDXs that have variable levels of stroma.

4) Tumours in Figure 1 should be better characterized in terms of size and proliferative index. I do not think that the + and – tumours differ in size at the time of resection, but this should be shown to ensure that increased metastasis is not due to more tumour volume.

5) The data in 2 D and E do not convincingly demonstrate differential fibroblast activation as loading seems uneven. This work should include gene expression data, replicates, a time course, as well as multiple fibroblast models.

6) Presentation of the proteomics data must be improved. The authors should show a PCA to demonstrate the extent to which the samples differ and to also determine variability in biological replicates. Differences between EV- and EV+ cargo should be displayed in a volcano plot that shows the fold difference of specific proteins as well as the significance of such differences when replicates are considered.

7) In figure 7, cutoffs should be indicated. Moreover, Tg2 should be correlated to the transcriptional signature of poorly migratory cells and/or stromal scores in order to validate the notion that it mediates the metastasis of this subset of cells through fibroblast activation.

*Reviewer #2 (Recommendations for the authors):*

1. Details of culture conditions for generation of MV should be provided in the text, and justified as to the in vivo tumor relevance of MV generation. In other words, are MV in vivo showing the same TGM2 relevant profile when comparing tumor types?

2. What do other MV genes show in KM plots of Figure 7b,c? Once again, does knockdown of TGM2 in the MV expts affect some of these other genes (e.g. MMP14) that have greater impact on survival?

3. Perhaps I missed this, but what is the effect of purified TGM2 enzyme compared to MV TGM2?

---

## [Author Response]

Essential revisions:1) The study should be extended to determine whether the results are cell-line specific. While focused on the TME, the study uses only a human cell-line derived xenograft model. At the very least, some of the experiments should be repeated in a syngeneic model (To account for immune cells and mouse-specific cytokine signaling) and/or an additional human model (PDX and/or cell line).

We agree with your concerns that the results may be cell-line specific. To address these concerns, we overexpressed Tg2 in a weakly migratory, weakly metastatic MCF7 breast cancer cell line. MCF7 do not express high levels of Tg2 and are very weakly migratory [Seo et al., “The GTP Binding Activity of Transglutaminase 2 Promotes Bone Metastasis of Breast Cancer Cells by Downregulating MicroRNA-205.”]. Overexpression of Tg2 in MCF7 cells did not significantly affect MCF7 single cell migration in 3D collagen (Supplemental Figure 5d-e). MVs isolated from MCF7 with Tg2 overexpression significantly increased fibroblast spreading compared to treatment with no MVs or MVs from MCF7 WT cells (Supplemental Figure 5f). Additionally, fibroblast traction force was significantly increased after culture with MCF7 + Tg2 MVs compared to control conditions (Supplemental Figure 5g). These findings highlight that MVs isolated from MCF7 with overexpression of Tg2 can induce a functional transformation of fibroblasts consistent with a myofibroblast-like phenotype. To determine whether MVs with Tg2 overexpression could increase MCF7 dissemination and outgrowth in spheroids, we used a spheroid coculture of MCF7 cells and fibroblasts. Treatment of spheroids with Tg2-overexpressing MCF7 MVs significantly increased spheroid outgrowth. MCF7 were visualized escaping the spheroid (Supplemental Figure 5h-i).

Importantly, it has been shown that the overexpression of Tg2 can increase metastasis of MCF7 (Seo et al., 2019, Am J Cancer Res). We injected MCF7 s.c. into NSG mice using the same breast cancer metastasis model used elsewhere in our study (Supplemental Figure 6a). During primary tumor growth, mice were injected s.c. at the primary tumor with either MCF7 MVs or MCF7+Tg2 MVs. Primary tumors were removed after 8 weeks, as tumor growth appeared to have plateaued. While no significant difference in tumor volume was quantified, the two largest tumors were from the MV group (Supplemental Figure 6b). From mice that were injected every 3 days with MCF7 MVs with overexpressed Tg2 compared to SF conditions, primary tumor *en bloc* sections were made and stained for anti-GFP. Notably, we observed higher numbers of GFP+ cells outside of the tumor margin suggesting that injection of Tg2-rich MVs increased cancer cell escape from the primary tumor. (Supplemental Figure 6c-d). Weekly bioluminescence imaging over 10 additional weeks revealed no indication of widespread metastasis. Mice were sacrificed 18 weeks after inoculation. After tissue collection, it was evident there was no robust metastasis to the liver, lungs, or lymph node. As metastasis was not seen in this model, these results suggest that while Tg2 can increase cancer cell escape from the primary tumor (Supp Figure 6 c-d), other components are required for cancer cells to complete all stages of metastasis (Supplemental Figure 6e).

Regarding the suggestion to use syngeneic and PDX models: We agree that syngeneic models are important as they feature full immunity and a natural stroma. However, difficulties in seeing robust metastasis in syngeneic models has been reported. Cyrus Ghajar’s group recently showed that the immune system can attack both GFP and luciferase labeling in cancer cells [Grzelak et al., “Elimination of Fluorescent Protein Immunogenicity Permits Modeling of Metastasis in Immune-Competent Settings.”]. Without GFP/luc expression, metastases cannot be monitored via BLI imaging and metastases at endpoint cannot be quantified with anti-GFP staining. As a result, monitoring metastasis in immunocompetent mouse models poses many difficulties. Ghajar’s group worked around this difficulty by engineering dendritic cells to produce GFP to induce tolerance to GFP expression. Importantly, this work by Ghajar’s group was just released in December 2021 after our paper was submitted. While their group and many other groups are currently working to generate successful cancer models in immunocompetent mice, it is beyond the scope to include in our study at this time. While the immune system is important for cancer progression, we also want to acknowledge that our study focuses on primary tumor mechanical remodeling.

In regard to PDX models, our work identifies a subpopulation of MDA-MB-231 breast cancer cells that are weakly migratory, yet capable of robust metastasis. To incorporate a PDX model, we would need to create a PDX from patients screened for weakly migratory cancer cells to investigate the role of Tg2, MVs, and fibroblast activation in their metastasis. Alternatively, we would need to sort patient-derived cells based on migration. Importantly, our transwell assay sorts cells over 20 rounds of sorting into highly and weakly migratory subpopulations which takes approximately 3 months. It is unclear if primary cells could withstand this prolonged in vitro sorting process. Technically, the likelihood of success for these experiments would be low and as such we did not pursue them.

2) Mechanistic studies should be enhanced. This could include a more thorough examination of CAF phenotypes or studies that better consider previous results using the same model.

CAF heterogeneity in cancer has garnered increased attention over the last few years. It is highly possible that MV signaling from heterogeneous populations of cancer cells leads to heterogeneous CAF phenotypes. The field of CAF heterogeneity is new and expanding and the current standard for detecting CAF subtypes is via single cell RNA sequencing. Additionally, the specific markers of myofibroblast-like CAFs vs immune regulatory-CAFs are highly dependent upon species, cancer type, and fibroblast origin. We chose to focus on functional changes to CAFs (spreading, proliferation, contractility, matrix deposition, etc.) as opposed to gene expression, as we feel that CAF function is ultimately the more important determinant for metastasis rather than their CAF phenotype determined solely by gene expression. Our work showing that MV-activated fibroblasts increase spreading, proliferation, contractility, and matrix deposition, in addition to using αSMA as a marker of CAF activation in vitro and in vivo, highly suggests that the phenotype of CAFs our study is focused on are myofibroblast-like. To further support this observation, we added a description to our Results section about how the fibroblast markers used in our studies fit within the current knowledge of CAF heterogeneity.

As our mice are immunocompromised, we believe that the effects of iCAFs on metastasis in vivo would be negligible. However, to address the comment, we completed qPCR on MDA^+^ and MDA^-^ tumors for a variety of iCAF markers (FSP1, IL-6, CXCL12) (see Author response image 1). qPCR analysis of MDA^+^ and MDA^-^ tumors for the mouse iCAF secreted factor IL-6 gene revealed similar stromal expression of IL-6. Additionally, expression of the iCAF-associated cytokine CXCL12 mouse gene was also similar in the MDA^+^ and MDA^-^ tumors. We additionally probed for FSP1 gene expression which is usually expressed on CAFs derived from epithelial or endothelial cells (rather than tissue-resident fibroblasts). FSP1+ CAFs can contribute to immune evasion through CCL2 production and can activate tumor immunity by promoting CD8^+^ T cell activation. While not significant, gene expression data showing a trend of increased in FSP1+ CAFs in the highly migratory MDA tumors. Taken together, our data suggests that MVs released from highly and weakly migratory cancer cells can induce different fibroblast phenotypes. We provide significant evidence that MV^—^ activated fibroblasts are myofibroblast-like (via spreading, proliferation, contractility, matrix deposition). As no difference in IL-6 or CXCL12 gene expression was detected in tumor stroma tissue, it is unlikely the MV^+^ induce a fibroblast phenotype associated with typical iCAF markers. In-depth analysis of fibroblast phenotypes would require single cell sequencing which we feel is outside the scope of the paper due to our focus on mechanical fibroblast signaling.

**Author response image 1. sa2fig1:** 

3) The authors should describe factors within MVs that may also mediate the observed effects. Also, the authors should address whether other proteins (including MMP-14) change with Tg-2 knockdown; and if these alterations may mediate the effects of Tg-2 on parameters such as survival. Finally, the effects of pure Tg-2 should be compared to the effects of Tg-2 containing-MVs.

We included a more thorough investigation into the proteomics data to determine what other factors in the MVs may induce fibroblast activation or matrix remodeling. Lists of “fibroblast-activating proteins” and “matrix remodeling proteins” were generated based on online datasets. All fibroblast-activating proteins tested were more highly expressed in MV^-^ compared to MV^+^, but TGM2 was the only protein on this list with significantly increased expression (Figure 3b-d).

A large variety of matrix-remodeling proteins were detected in the MV proteomics, including matrix ligands, proteases, protease inhibitors, and crosslinking enzymes. Interestingly, MV^+^ had significantly higher levels of the matrix remodeling proteins TIMP3, FN1, and COL8A1 (Figure 3d). MV^-^ had significantly higher levels of the crosslinking enzymes PLOD1 and PLOD3, the matrix ligand COL12A1, and TGM2 (Figure 3d). As TGM2 can be categorized as both a matrix remodeling and fibroblast-activating protein and was significantly greater in the MV^-^ compared to MV^+^, we believe this addition to the paper reinforces our focus on TGM2 (Figure 3).

We additionally investigated whether other proteins in MVs change with Tg2 knockdown. To address the concerns, we completed proteomics analysis comparing a scrambled control and Tg2 knockdown MV^-^ (Supplemental Figure 3h-k). We discuss the results of this analysis in the paper section titled “Modulation of MV-Tg2 expression regulates MV-mediated fibroblast activation and cancer cell dissemination in vitro”

We additionally investigated whether purified Tg2 was capable of inducing fibroblast transformation (Supplemental Figure 5j-k). Treatment of fibroblasts with 1 ug/mL of purified Tg2 resulted in significantly increased fibroblast spreading and fibroblast traction force compared to control conditions. Additionally, the levels of fibroblast transformation seem to be comparable to levels induced by MV^-^.

4) Presentation of the proteomics data must be improved. The authors should show a PCA to demonstrate the extent to which the samples differ and to also determine variability in biological replicates. Differences between EV- and EV+ cargo should be displayed in a volcano plot that shows the fold difference of specific proteins as well as the significance of such differences when replicates are considered.

In response to comment 3 (above) and comment 4, we reworked our presentation of the proteomics data (Figure 3b-d). We changed our presentation to volcano plots and added the p=0.01 significant cut-off line. Thank you for this suggestion and we agree that the data is much easier to visually interpret now.

We are unable to generate a PCA as only 1 replicate of iTRAQ proteomics was completed. A benefit of iTRAQ proteomics is that statistics can be generated from 1 replicate. Because we additionally verified with a western blot that the differences in Tg2 between the subpopulations are robust in vitro, we do not feel that the expense of completing 3 replicates of proteomics is justifiable.

5) The data in 2 D and E do not convincingly demonstrate differential fibroblast activation as loading seems uneven. This work should include gene expression data, replicates, and a time course.

Thank you for bringing this to our attention. We have revised Figure 2E (now Figure 2F) with a new western blot with equal loading. We revised our manuscript to include quantification of western blots with N=3+ replicates to further support our conclusions (Figure 2e, 2g, 3f, 3h, Supp 4b, Supp 4g, Figure 3c, 3h)

We show that FAK activation can occur in 30 minutes (Figure 2D) while aSMA expression is evident 24 hours after MV treatment (Figure 2F). It may be interesting for future research to investigate how long the effects of MV activation can last (and whether some type of feed forward loop exists in fibroblasts to further induce activation). However, we believe this to be outside the scope of this study.

Unfortunately, the culture system used for MV-fibroblast cocultures presents technical challenges for RNA isolation for gene expression studies. For our studies, fibroblasts are cultured on 20 kPa polyacrylamide gels in single cell conditions. It is well-documented that culture of fibroblasts (specifically NIH 3T3) to confluency will induce a myofibroblast-like phenotype. To address the concerns for gene expression data, we scaled up our PA gel size from 22 x 22 mm in area to 45 x 56 mm in area. However, we were still unable to isolate sufficient RNA for qPCR experiments. Lastly, it is important to note that pFAK cannot be detected using qPCR as phosphorylation occurs at the protein level. To further support our western blots and address the concerns, we included immunofluorescence staining of fibroblasts cultured on PA gels for aSMA (Supp. Figure 1a, Supp. Figure 4a).

6) A serum supplemented control should be used for the in vivo experiments conducted in Figure 6, to better match the experimental conditions.

Thank you for bringing this concern to our attention. We realize that our wording was not clear. MVs are isolated under serum-free conditions and after isolation are resuspended in serum-free media. For this experiment, our mice were injected with either MVs suspended in serum-free media or serum-free media alone. We have revised the text to explain this more thoroughly.

7) Details of culture conditions for generation of MV should be provided in the text, and justified as to the in vivo tumor relevance of MV generation.

We revised our manuscript to include more details on MV isolation in vitro in both the results and method sections. We appreciate your concern over the relevance of MV generation in vivo compared to the in vitro system utilized to isolate MVs. Importantly, isolating EVs in vivo is technically challenging. If isolating from the blood, limitations in the amount of blood that can be isolated from mice highly limit the number of EVs that can be isolated. While new techniques are emerging to isolate EVs from primary tumors, the EV population is highly heterogeneous and will contain EVs from all cell types, not just cancer cells. In order to try to recapitulate a cell-dense primary tumor, we isolate EVs from the conditioned media of ~70% confluent cancer cells. This is the current standard for in vitro EV isolation. Many groups are working on expanding the throughput of EV isolation in vitro through the use of bioreactors. However, this is an emerging field and research is still being conducted to determine the differences between MVs isolated from cells on tissue culture plastic, in bioreactors, in 3D in vitro *systems*, or in vivo. We agree that these findings will be very important for the future of EV research but are not yet at a point where they can be included in this study.

8) In figure 7, cutoffs should be indicated. Moreover, Tg-2 should be correlated to the transcriptional signature of poorly migratory cells and/or stromal scores in order to validate the notion that it mediates the metastasis of this subset of cells through fibroblast activation.

We have revised the figure to include cut-offs and they are now labeled in the figure. We additionally used patient data to look at the correlation between Tg2 expression and fibroblast activation (Figure 7D). Tg2 expression in breast cancer patients was significantly correlated with expression of 4 fibroblast markers, ACTA2, FAP, PDGFRa, and PDGFRb. We believe this result greatly strengthens our in vitro and in vivo data highlighting the relationship between Tg2 and fibroblast activation.

We appreciate your comment about comparing the transcriptional signature of weakly migratory cells to Tg2 expression. Importantly, it is not possible to find a gene set for highly metastatic yet weakly migratory cancer cells outside of our phenotypic sorting. One of the most interesting findings in our work is that within a highly mesenchymal and migratory cell line (MDA-MB-231), there are weakly migratory cells that seem to be highly metastatic. To address your concern, we utilized RNA sequencing of our highly and weakly migratory breast cancer cells. Using genes involved in stromal-activation, we scored MDA^+^ and MDA^-^ based on their potential to activate the stroma (through inflammatory cytokines, matrix ligands, cell-cell adhesion proteins, and secreted factors) (see Figure 1k-q). Interestingly, when all genes were grouped together, MDA^+^ and MDA^-^ seemed to have comparable stromal-activating scores (Figure 1I).

Importantly, this does not match the fibroblast activation quantified in vivo. Therefore, we grouped genes into 4 stromal-activating groups: inflammatory, matrix ligands, cell-cell adhesion, and secreted. Within the secreted group, TGFB2 and TGM2 were more highly expressed by MDA^-^ compared to MDA^-^ suggesting that factors secreted by MDA^-^, such as TGFB2 or TGM2, may contribute to the stromal differences observed in vivo.

Reviewer #1 (Recommendations for the authors):The current manuscript by Schwager and colleagues describes a mechanism by which poorly migratory MDA-MB-231 cells can be metastatic. This study follows a recent paper from the same group (published in January) demonstrating that these poorly migratory cells are more metastatic than their highly migratory counterparts, and that this is due at least in part to E-Cadherin expression and the ability to form circulating tumour cell (CTC) clusters. In the current study, the authors show that the low migratory cells secrete unique EVs that can activate fibroblasts, concomitant with metastatic progression, and that this function is dependent on the presence of Tg-2. The novelty of this work is in the phenotypic heterogeneity of tumour cells, even within cell lines, and the importance the microenvironment in mediating metastasis associated with this diversity. While interesting, this work uses only one model, which was very recently published. The study, I think, would require repetition within additional models, as well as the inclusion of mechanistic studies designed to determine why the EV cargo differs between the highly and poorly migratory subclones.1) Fibroblasts are also heterogeneous. CAF populations in migratory versus poorly migratory tumours should be more thoroughly explored to account for the possibility that CAF subtypes change between conditions and in response to EVs.

Please refer to “Essential Revision #2”.

2) The current study does not really reconcile the mechanisms revealed in the published paper that uses the same model. For example, it does not attempt to compare RNAseq data with proteomics data or to determine whether EVs affect E-Cadherin expression. It also does not explore whether CAFs may be needed for the formation of CTC clusters.

Thank you for this interesting comment. We agree that the relationship between EVs and EMT status is very interesting. We additionally think that CAF-cancer cell CTC clusters are very interesting and EVs from both CAFs and cancer cells may play a functional role in this process. However, we feel that both of these topics are outside the scope of this paper. While E-cadherin and EMT status were the focus of our 2021 paper in Cancer Research, our paper here serves to determine how weakly migratory cancer cells are capable of escaping the primary tumor when they are weakly migratory. We do not focus on the role of E-cadherin or the relationship between clustering and survival in the vasculature in this study.

We have added a figure comparing RNASeq of MDA^+^ and MDA^-^ with proteomics of MV^+^ and MV^-^ to determine whether the gene signatures observed in cells is consistent with the protein differences seen in MVs (Supplemental Figure 2d). The majority of genes/proteins identified in both data sets were not significantly different between MDA^+^ vs MDA^-^ and MV^+^ vs MV^-^, highlighting the overall genomic/proteomics similarity between these two phenotypically distinct subpopulations. This was expected as these cells and their MVs are derived from the same cell line (MDA-MB-231). This finding also highlights that MVs highly reflect the composition of the cell they were released from.

3) While focused on the TME, the study uses only a human cell-line derived xenograft model. At the very least, the work should be repeated in a syngeneic model (To account for immune cells and mouse-specific cytokine signaling) as well as several more human models, ideally including PDXs that have variable levels of stroma.

Please refer to “Essential Revision #1”.

4) Tumours in Figure 1 should be better characterized in terms of size and proliferative index. I do not think that the + and – tumours differ in size at the time of resection, but this should be shown to ensure that increased metastasis is not due to more tumour volume.

We added a growth curve to Figure 1 (Figure 1C) demonstrating that the two cells have similar growth rates. We also added a statement citing our Cancer Research paper which characterizes primary tumor volume and cell Ki67 status.

5) The data in 2 D and E do not convincingly demonstrate differential fibroblast activation as loading seems uneven. This work should include gene expression data, replicates, a time course, as well as multiple fibroblast models.

Please refer to “Essential Revision #5”.

6) Presentation of the proteomics data must be improved. The authors should show a PCA to demonstrate the extent to which the samples differ and to also determine variability in biological replicates. Differences between EV- and EV+ cargo should be displayed in a volcano plot that shows the fold difference of specific proteins as well as the significance of such differences when replicates are considered.

Please refer to “Essential Revision #4”.

7) In figure 7, cutoffs should be indicated. Moreover, Tg2 should be correlated to the transcriptional signature of poorly migratory cells and/or stromal scores in order to validate the notion that it mediates the metastasis of this subset of cells through fibroblast activation.

Please refer to “Essential Revision #8”.

Reviewer #2 (Recommendations for the authors):1. Details of culture conditions for generation of MV should be provided in the text, and justified as to the in vivo tumor relevance of MV generation. In other words, are MV in vivo showing the same TGM2 relevant profile when comparing tumor types?

Please refer to “Essential Revision #7”.

2. What do other MV genes show in KM plots of Figure 7b,c? Once again, does knockdown of TGM2 in the MV expts affect some of these other genes (e.g. MMP14) that have greater impact on survival?

Please refer to “Essential Revision #3”

We additionally investigated whether other proteins in MVs change with Tg2 knockdown. To address the concerns, we completed proteomics analysis comparing a scrambled control and Tg2 knockdown MV^-^ (Supplemental Figure 3h-k). We discuss the results of this analysis in the paper section titled “Modulation of MV-Tg2 expression regulates MV-mediated fibroblast activation and cancer cell dissemination in vitro”

Additionally, KM plots for COL12A1, PLOD1, and PLOD3 (the other fibroblast-activating/matrix remodeling protein significantly enriched in MV^-^) were included in the supplement (Supplemental Figure 7).

3. Perhaps I missed this, but what is the effect of purified TGM2 enzyme compared to MV TGM2?

Please refer to “Essential Revision #3”